

# Single-footprint retrievals of temperature, water vapor and cloud properties from AIRS

Fredrick W. Irion[1], Brian H. Kahn[1], Mathias M. Schreier[1], Eric J. Fetzer[1], Evan Fishbein[1], Dejian Fu[1], Peter Kalmus[1], R. Chris Wilson[2], Sun Wong[1], and Qing Yue[1]

[1]Jet Propulsion Laboratory, California Institute of Technology, Pasadena, CA 91106, USA.
[2]Joint Institute for Regional Earth System Science and Engineering, University of California, Los Angeles, CA 90095, USA.

*Correspondence to*: Fredrick W. Irion (bill.irion@jpl.nasa.gov)

**Abstract.** Single-footprint Atmospheric Infrared Sounder spectra are used in an optimal estimation-based algorithm (AIRS-OE) for simultaneous retrieval of atmospheric temperature, water vapor, surface temperature, cloud-top temperature, effective cloud optical depth and effective cloud particle radius. In a departure from currently operational AIRS retrievals (AIRS-V6), cloud scattering and absorption are in the radiative transfer forward model, and Level 1b AIRS thermal infrared data are used directly rather than Level 2 cloud-cleared spectra. Coincident MODIS Level 2 cloud data are used for cloud a priori. Using Level 1b spectra improves the horizontal resolution of the AIRS retrieval from ~45 km to ~13.5 km at nadir, but as microwave data are not used, retrieval is not made at altitudes below thick clouds. An outline of the AIRS-OE retrieval procedure and information content analysis is presented. Initial comparison of AIRS-OE to AIRS-V6 results show increased horizontal detail in the water vapor and relative humidity fields in the free troposphere above clouds. Comparisions of temperature, water vapor and relative humidity profiles against coincident radiosondes show good agreement. Future improvements to the retrieval algorithm, and to the forward model in particular, are discussed.

## 1 Introduction

The advantage of hyperspectral nadir measurement in the thermal infrared over the microwave is higher vertical resolution of retrieved temperature and water vapor. Operational instruments such as the Atmospheric Infrared Sounder on the EOS Aqua platform (AIRS; Aumann et al., 2003), the Infrared Atmospheric Sounding Instruments on Metop-A and –B (IASI; Blumstein et al., 2004), and the Cross-track Infrared Sounder on Suomi NPP (CrIS; Han et al., 2013) provide global radiance data for assimilation into weather forecasting and reanalysis models, and profile retrievals for process studies. Perhaps the largest complication for global retrievals (and assimilation) using infrared spectra is near-ubiquitous, highly variable cloud absorption and scattering in the instrument field-of-view (FOV). This is illustrated in Figure 1, which shows sample brightness temperature spectra observed by AIRS in nine adjacent footprints. Most of the variation between the spectra is from differences in cloud-top temperature and cloud optical depth, and to a lesser extent, cloud particle radius as seen on the AIRS footprint. Two general approaches have been used in obtaining profile retrievals from cloudy infrared spectra. The first has been "cloud-clearing," where the temperature and trace gas fields (including water vapor) are treated as constant across adjacent thermal infrared footprints, and only the cloud field varies. A Level 2 cloud-free infrared spectrum is calculated from these cloudy Level 1 spectra, and then used for profile retrievals over a larger field-of-regard (FOR) (e.g., Susskind et al., 2011.) Cloud-clearing simplifies calculations as the forward model does not incorporate scattering or absorption by clouds. However, by combining retrieval footprints and assuming constant non-cloud quantities, cloud-clearing can mask significant horizontal gradients, particularly in water vapor, which can have a much shorter horizontal length scale than temperature especially in low- and midlatitudes (e.g., Kahn and Teixeira, 2009).





The other approach that accounts for clouds is to include cloud absorption and scattering in a retrieval forward model. Advances in efficient cloud scattering algorithms and ever-increasing computing power hold the promise of incorporating explicit cloud effects in forward models for routine, operational retrievals. Several methods and software packages have been developed for the calculation of outgoing radiance in the presence of clouds: Optimal Spectral Sampling (OSS; Moncet et al., 2015), The

Havermann-Taylor Fast Radiative Transfer Code (HT-FRTC; Havermann 2006), The Principal Component-based Radiative Transfer Model (PCRTM; Liu, 2005), Discrete Ordinate Algorithm for Radiative Transfer (DISORT; Laszlo et al., 2016), Vector Linearized Discrete Ordinate Radiative Transfer (VLIDORT; Spurr, 2006), the Community Radiative Transfer Model (CRTM; Han et al., 2006), the Radiative Transfer for the Television Infrared Observation Satellite (TIROS) Operational Vertical Sounder algorithm (RTTOV; Saunders et al., 2013), and, as used in this work, the Stand-alone AIRS Radiative Transfer Algorithm +

Delta-Four Stream (SARTA-D4S; Ou et al., 2013).

While incorporating cloud scattering usually makes for a more complicated and computationally expensive radiative transfer calculation, the horizontal resolution of retrieved species can be improved compared to cloud-cleared results. Several methods have been employed for direct use of cloudy infrared spectra in atmospheric retrievals. Among them are combining channel radiances into "super-channels" using empirical orthogonal functions (Liu et al., 2009), neural networks (Blackwell, 2005), and

parameterization of frequency-dependent non-scattering optical depths (Kulawik et al., 2006a). Here we describe a new retrieval scheme, Optimal Estimation Retrieval System for AIRS ("AIRS-OE") that can use the L1b radiances of single AIRS footprints, without cloud-clearing, for the retrieval of temperature profiles ($T_{atm}$), $H_2O$ volume mixing ratio profiles, skin temperatures ($T_{sfc}$), effective cloud optical depths over an AIRS field-of-view ($\tau_{eff}$), cloud-top temperatures ($T_{cldtop}$) and effective particle radii ($r_{eff}$). $CO_2$ and $O_3$ profiles, while not the primary constituents examined or validated here, are also retrieved to improve spectral fitting

and temperature results. The goal is to improve the horizontal resolution of retrievals by using the less-processed Level 1b AIRS infrared radiance (at ~13.5 km nadir horizontal resolution), rather than the ~45 km resolution Level 2 cloud-cleared radiance produced by the currently-operational Version 6 ("AIRS-V6") retrieval algorithm (Susskind et al., 2003; 2014.) Additional differences of AIRS-OE from the current AIRS-V6 retrieval include:

(1) Cloud optical depth, effective particle radius, and cloud-top temperature are now explicitly in the forward model, and

are retrieved along with temperature and water vapor profiles. A priori cloud information for cloud retrieval is calculated from the MYD06 dataset of the Moderate Resolution Imaging Spectroradiometer (MODIS) instrument, co-located on the EOS-Aqua platform (Platnick et al., 2003, 2017).

(2) For a priori temperature profiles, skin temperature and water vapor profiles, we use European Center for Medium-Range Weather Forecasting (ECMWF) 6-hour analyses linearly interpolated by time and space to the AIRS observation,

where the AIRS-V6 retrieval uses a neural network trained on AIRS radiances and ECMWF re-analyses (Blackwell, 2005).

(3) In a departure from the singular-value-decomposition technique of AIRS-V6, which does not use an a priori covariance, the new retrieval uses an optimal estimation scheme (e.g., Rodgers, 2000), similar to that of the nadir-sounding Tropospheric Emission Sounder (TES; Bowman et al., 2006).

(4) Aside from an initial retrieval of the cloud optical depth, retrievals of atmospheric constituents are made simultaneously, rather than sequentially as in AIRS-V6.



(5) AIRS-OE uses only the thermal infrared data from AIRS, and does not use ~45 km nadir resolution microwave data from the co-located Advanced Microwave Sounding Unit (AMSU). Profiles cannot be retrieved below IR-opaque clouds effectively covering a pixel. However, unlike AIRS-V6, temperature, water vapor etc. are not assumed to be uniform across the nine AIRS footprints in an AMSU field-of-regard (FOR).

In this paper, we give a very brief overview of the AIRS and MODIS instruments, and outline the retrieval and the information content analysis. Some sample cloud property results are presented, and we show an intial comparison with near-coincident measurements by Cloudsat/CALIPSO. We then compare temperature, water vapor and relative humidity results to those of the operational AIRS-V6 retrieval, and with sets of near-coincident high quality radiosondes.

## 2 Instrument and Spectral Data

The Atmospheric Infrared Sounder (AIRS) instrument is a thermal infrared grating spectrometer, with 2378 channels between 3.7 and 15.4 μm.  In sun-synchronous, polar orbit on the EOS Aqua satellite, AIRS delivers approximately 2.9 million spectral observations every 24 hours. AIRS was designed for co-located measurements with the Advanced Microwave Sounding Unit (AMSU) microwave instrument, with nine AIRS observations (each with nadir horizontal resolution of ~13.5 km) in a 3 x 3 grid over a single AMSU observation with a nadir horizontal resolution of ~45 km (Aumann et al., 2003). AIRS was designed to

provide global data on weather and climate processes, and is a key antecedent to the IASI and CrIS spectrometers.

As noted, operational Level 2 data from the Moderate Resolution Imaging Spectroradiometer (MODIS) are used as a priori for cloud-top temperature, and during daytime observation, cloud optical depth and particle radius. Co-located with AIRS on EOS-Aqua, MODIS observes in 36 spectral bands from 0.4 to 14.4 μm with horizontal resolutions (depending on band) ranging from 250 m to 1 km at nadir. Details about the MODIS cloud optical properties in the MYD06 data set are found in Platnick et al.

(2017). Prior to use, the MODIS cloud data are mapped and weighted over the AIRS footprint as described below in Sec. 3.2.2.

## 3 AIRS-OE Retrieval overview

Figure 2 shows a simplified block diagram of the AIRS-OE retrieval procedure. For convenience to the reader, blocks are annotated with the Section numbers of this paper where a more detailed description can be found. Input parameters are described below in Sect. 3.1 through 3.5. A brief description of the retrieval itself is in Sect. 3.6, followed by information content analyses

(3.7) and quality control filters (3.8).

### 3.1 Optimal estimation cost function

The mathematical basis for optimal estimation retrievals is described by Rodgers (2000). Implementation is similar to that of TES (Bowman et al, 2006) with significant differences in the treatment of clouds. The retrieval algorithm minimizes the difference between an observed and a forward-modelled radiance, subject to a quadratic constraint, through the cost function:

$C = (\mathbf{y} - \mathbf{F}(\hat{\mathbf{x}}, \mathbf{b}))\mathbf{S}_\varepsilon^{-1}(\mathbf{y} - \mathbf{F}(\hat{\mathbf{x}}, \mathbf{b}))^{-1} + (\hat{\mathbf{z}} - \mathbf{z}_\mathrm{a})\mathbf{S}_a^{-1}(\hat{\mathbf{z}} - \mathbf{z}_\mathrm{a})^{-1}$     (1)

where:

   $\mathbf{y}$ is the vector of measured radiances,

   $\mathbf{F}(\hat{\mathbf{x}}, \mathbf{b})$ is the forward-model radiance,

   $\hat{\mathbf{x}}$ is the "full" state vector, described below,

$\mathbf{b}$ contains additional variables needed (but not retrieved) and observational metadata (e.g., scan angle) for calculating radiances,

   $\hat{\mathbf{z}}$ is the "retrieval" state vector, described below,





$\mathbf{z}_a$ is the a priori retrieval state vector,

$\mathbf{S}_\varepsilon^{-1}$ is the inverse radiance noise covariance, and

$\mathbf{S}_a^{-1}$ is the inverse covariance of the a priori $\mathbf{z}_a$.

(Note that we accent a retrieved quantity with a caret, e.g., $\hat{\mathbf{z}}$, to distinguish it from the "true" quantity, $\mathbf{z}$.) The a priori state
vector, $\mathbf{z}_a$, is also the first guess in a retrieval, except where noted below for cloud optical depth. The full state vector, $\hat{\mathbf{x}}$, has as
many elements for each retrieved profile constituent ($T_{atm}$, $H_2O$, $O_3$ and $CO_2$) as there are layers in the forward model at or above
ground, plus those for retrieved scalar quantities, $T_{sfc}$, $T_{cldtop}$, $\tau_{cld}$ and $r_{eff}$. There are a maximum of 100 layers filled in our forward
model from the surface upwards, on its fixed pressure grid with level pressures from 1100 to 0.1 mb (as described in Strow et al.,
2003). The pressure layers are constructed as the log-mean of the upper and lower pressures levels:

$$P_{layer} = \frac{P_2 - P_1}{ln(P_2/P_1)} \qquad (2)$$

Following Bowman et al. (2006), the forward-model layer gridding must be fine enough for calculation of the observed radiance,
but is usually much finer than the vertical resolution of a retrieved profile. The retrieval state vector, $\mathbf{z}$, has a reduced number of
layers, which varies by constituent, to reflect a lower vertical resolution. (A maximum 42 layers are retrieved for $T_{atm}$, 28 for
$H_2O$, 10 for $CO_2$ and 9 for $O_3$.) The retrieved state vector, $\hat{\mathbf{z}}$, is mapped to the full state vector, $\hat{\mathbf{x}}$ when the forward model is
called to calculate a radiance:

$$\hat{\mathbf{x}} = \mathbf{M}\hat{\mathbf{z}} \qquad (3)$$

In our retrieval, the matrix $\mathbf{M}$ performs a piecewise linear interpolation by log pressure from the lower number of layers in $\hat{\mathbf{z}}$ to
the higher number of layers in $\hat{\mathbf{x}}$. For gas profiles, $\tau_{cld}$, and $r_{eff}$, logarithmic quantities are used in the state vector to ensure that
their linear values always remain positive as input to the forward model during retrieval iterations. Retrievals for $T_{atm}$, $T_{sfc}$, and
$T_{cldtop}$ are linear quantities. The state vector thus usually contains both linear and logarithmic elements. Description and
determination of the different elements of the cost function are described in the sections below.

### 3.2 A priori information

#### 3.2.1 Temperature profile, water vapor, surface temperature, ozone and carbon dioxide

Initial guess profiles for $T_{atm}$, $H_2O$, $T_{sfc}$, and surface pressure (the latter remaining fixed during the retrieval) are derived from
ECMWF analysis data, at 0.25° and 6 hour resolution, linearly interpolated in time and space to that of the observed footprint,
with vertical profiles linearly interpolated by the logarithm of the retrieval pressure gridding. Initial $O_3$ profiles are calculated
from the climatology of McPeters et al. (2007). A priori $CO_2$ profiles are calculated by formulae developed by G. C. Toon
(personal communication), and are similar to those used by the Total Carbon Column Observing Network (Wunch et al., 2011).

#### 3.2.2 MODIS cloud a priori information and mapping to AIRS footprint

For a priori data on $T_{cldtop}$, $\tau_{cld}$ and $r_{eff}$ in each AIRS observation, weighted averages of MODIS Level 2 data are made over the
AIRS spatial response function (SpatialRF). An average AIRS SpatialRF is first calculated for the observation (see Schreier et
al., 2010). The SpatialRF varies by scan angle and spectral channel, and we make a simple average of the SpatialRF using only
the channels used in the retrieval. Using formulae described by Pagano et al. (2015), the spatial response is mapped over the
target scene with an approximate resolution of 0.5 km at nadir. MODIS data from the MYD06_L2 (Aqua) product are then
mapped in the vicinity of the AIRS observation. These data have a horizontal resolution of about 1 km at nadir. The mapped
AIRS SpatialRF is then spatially interpolated onto this MODIS horizontal mapping, and normalized to sum to unity.




Once the AIRS and MODIS footprints are colocated, MODIS retrieval fields for (1 km) cloud-top temperature, cloud optical thickness, and cloud effective radius are extracted and mapped (as data are available). Figure 3 illustrates a sample MODIS $T_{cldtop}$, $\tau_{cld}$ and $r_{eff}$ field overlaid by the AIRS SpatialRF. Weighted averages and weighted standard deviations of the MODIS $T_{cldtop}$ and $r_{eff}$ are then calculated on the interpolated AIRS SpatialRF, excluding MODIS cloud-free pixels. Calculations are

similar for $\tau_{cld}$, except cloud-free pixels are included in the averaging calculation as having an optical depth of zero. From the MODIS Cloud_Mask_1km field, we extract and similarly map the cloud mask status (0 = undetermined, 1 = determined), cloud mask cloudiness (0 = confidently cloudy or fill if status flag = 0, 1 = probably cloudy, 2 = probably clear, 3 = confidently clear) and thin cirrus flags (0 = yes or fill if status flag = 0, 1 = no). The weighted averages of these flags over the AIRS scene are used to decide (a) if the scene is clear, (b) a cirrus cloud too thin for a confident MODIS retrieval of its cloud-top temperature is in the

scene, but retrieval by AIRS should be attempted using an assumed cloud-top temperature a priori, or (c) retrieval in a cloudy scene will be attempted using the averaged MODIS cloud data as a priori. We categorize AIRS scenes as clear, thin cirrus, or cloudy (noting that MODIS does not report $\tau_{cld}$ and $r_{eff}$ at night), and set a priori cloud data accordingly. We describe the criteria for these bins in turn, and the cloud a priori selected for them.

*Clear*

An AIRS scene is treated as clear if

     (a)   the average for the Cloud Mask Status Flag is greater than 0.95, and

     (b)   the average Cloud Mask Cloudiness is greater than 2.5, and

     (c)   the average Thin Cirrus Flag is greater than 0.9.

In this case, no cloud information is in the retrieval or full state vector, and the retrieval algorithm goes directly to retrieve $T_{atm}$,

$T_{sfc}$, $H_2O$, $O_3$ and $CO_2$.

*Thin cirrus daytime and nightime*

A scene is considered to have thin cirrus but of unknown temperature if the average Thin Cirrus Flag is ≤ 0.9 and either the following are true:

     (a) Cloud-top temperature cannot be calculated because of missing values, or

(b) both average cloud mask status > 0.95 and Cloud Mask Cloudiness > 2.8.

In this case, we assume a default a priori $T_{cldtop}$ of 230K. For daytime scenes, the initial $\tau_{cld}$ is set to 0.1. For nightime, the initial optical depth thickness is set to 1. For both day and night, we set the initial $r_{eff}$ at 40 µm.

*Cloudy scenes daytime and nighttime*

A scene is treated as cloudy (ice or water cloud) if the weighted average of the MODIS cloud-top temperature on the AIRS

footprint can be calculated. The result is used as the a priori $T_{cldtop}$. If $\tau_{cld}$ can be calculated, it is used as the a priori, but is set to no less than $1 \times 10^{-3}$. If a $\tau_{cld}$ cannot be calculated during daytime, then the a priori $\tau_{cld}$ is set to $1 \times 10^{-3}$. (This lower limit was set to examine retrieval sensitivity for $\tau_{cld}$, which can depend not only on the true $\tau_{cld}$ itself, but also the thermal contrast with the ground. Initial tests indicate that the lower limit for any sensitivity to $\tau_{cld}$ is ~0.005.) At nighttime, the a priori $\tau_{cld}$ is 1. If the weighted average of the MODIS $r_{eff}$ can be calculated, it is used as the a priori. If the average cannot be calculated, or if it is

nighttime, the default $r_{eff}$ is 40 µm. However, any reported particle radii outside the range of from 5 to 85 µm use extrapolated cloud absorption and scattering parameters, and may not be reliable.



### 3.2.3 Emissivity

Wavelength-dependent surface emissivities are input as fixed parameters, and are not retrieved or modified; this may be revised in future versions of AIRS-OE. For ocean emissivity, we use the National Center for Environmental Prediction – Environmental Modeling Center (NCEP/EMC) Infrared Sea Surface Emissivity (IRSSE) formulae and coefficients (van Delst, 2003), calculated

for channel frequency, view angle and wind speed, with the latter estimated from the ECMWF analysis (described above). Land emissivity is from the Cooperative Institute for Meteorological Satellite Studies, University of Wisconsin - Madison Global Infrared Land Surface Emissivity Database (Seemann et al., 2007). These are monthly maps of land emissivity at 10 wavelengths from 3.6 to 14.3 μm, gridded spatially by 0.05°. This spatial gridding is smaller than the AIRS footprint, so we spatially interpolate the emissivity to the coincident MODIS gridding (described in Sect. 3.1.2 and as illustrated in Fig. 3). If the target

scene is a mixed land/ocean surface, we calculate the ocean emissivities (as above) and use them to fill in the ocean parts of the MODIS grid. With the emissivities at the 10 wavelengths on the MODIS gridding, the AIRS SpatialRF is used to calculate weighted averages. Emissivities for each AIRS retrieval channel are then calculated by interpolation by wavelength from these weighted averages. As the emissivitity database does not extend prior to calendar year 2003, we (arbitrarily) use the data from 2003 for observations made in 2002.

### 3.3 A priori covariances

In this initial evaluation of the retrieval described here, a priori covariances ($S_a$ in Eq. 1) are listed in Table 2. The covariances are ad hoc, but guided by previous experience with AIRS and TES retrievals. We recommend caution in applying resultant errors, although they may still be useful in comparing results between retrievals. Note that $H_2O$, $O_3$, $CO_2$, $\tau_{cld}$ and $r_{eff}$ are retrieved as $\log_e$ quantities, and the covariances of their logarithms are used.

Off-diagonal elements of the covariance matrices are created using assumed length scales:

$$\sigma_{ij}^2 = \sigma_i \sigma_j \exp\left(-\frac{|z_i - z_j|}{l}\right)$$

(4)

where:

$\sigma_{ij}^2$ is the off-diagonal covariance for layers $i$ and $j$,

$\sigma_i, \sigma_j$ are the square roots of the on-diagonal covariances,

$z_i, z_j$ are the estimated altitudes, and

$l$ is the assumed length scale.

The off-diagonal length scale for temperature and water vapor was kept low (0.5 km) as this tended to reduce unrealistic retrievals at layers below clouds from adversely affecting retrievals above clouds. (This is discussed further in Sect. 3.9 below.) Covariance matrices are calculated individually for each constituent and then "stacked" into a larger matrix for use in the

simultaneous retrieval. At present, we are not using covariances between constituents (say, between temperature and water vapor), but this will be investigated for use in later versions.

### 3.4 Forward model

The forward model is the Standalone AIRS Radiative Transfer Algorithm (SARTA) (Strow et al., 2003; 2006), supplemented with a delta-four-stream (D4S) calculation for cloud transmissivity (Ou et al., 2013). This joint SARTA+D4S forward model has





been used to retrieve ice cloud parameters from single-footprint AIRS observations (Kahn et al., 2014), and of three forward models tested, using SARTA+D4S produced the lowest biases in temperature and water vapor compared to coincident sondes in clear scenes. Here, we use this model to additionally retrieve water cloud properties. For ice clouds, scattering parameters are from Baum et al. (2007). For water clouds, we use Mie scattering parameters calculated using formulae from Mishchenko et al.

5 (2002).

Within the SARTA+D4S forward model, SARTA calculates the cloud-free gaseous transmissivities for each pressure layer and retrieval channel given the temperature and gas profiles, emissivity, scan angle, etc. The D4S code calculates the cloud transmissivities for retrieval channels given a cloud-top temperature, optical depth and particle size. As each pressure layer in the forward model is assumed homogeneous, the gaseous transmissivity for each channel is multiplied with that of the cloud to

produce a combined transmissivity within a single layer. Note that this assumes the cloud can be modelled to "fit" in one vertical layer, no matter how thick the cloud. The forward model layer selected for this is the tropospheric layer, lowest in pressure, with an atmospheric temperature higher than or equal to the cloud-top temperature (or the next lower-pressure layer if there the absolute difference between the cloud-top temperature and the layer temperature is less).

Only cirrus parameters are used at cloud-top temperatures below 253.15 K, while only Mie cloud parameters are used at

temperatures above 273.15 K. Between these temperatures, we use a sliding weight between Mie and cirrus-derived cloud. This approach may overestimate the amount of ice occurrence as the majority of cloud tops within the temperature range of mixed phase and supercooled clouds (233 - 273 K) is mostly liquid according to lidar estimates of phase (e.g., Hu et al., 2010). This transition of phase was tested by changing the lower boundary of this "mixed phase" range in the retrieval from 253.15 K to 233.15 K, and reprocessing a test granule (#44, Sept. 6, 2002, described below). Changes were scattered and isolated in retrieved

$T_{atm}$, $H_2O$ and relative humidity (the calculation of which is described in Sec. 3.7.3), and were largest in the boundary layer, with about 6% of retrieved relative humidities changing by more than ±5% at ~900 mb. For clouds where the MODIS-derived a priori cloud-top temperature was less than 273.15 K, about 9% of $T_{cldtop}$ changed by more than ±5 K, and 21% of $\tau_{cld}$ changed by more than ±10%. However, these changes tended to happen in regions where there was a high standard deviation in the MODIS-derived average cloud-top temperature over the AIRS FOV (> 20 K), which is more likely to contain a mixture of ice-only,

liquid-only and mixed-phase clouds. Put simply, the clouds in such scenes are more complicated to model, and more investigation is warranted.

This forward modelling of temperature, trace gases and cloud properties, while computationally fast, is best suited for optically and geometrically thin clouds, and may not be well suited for thick clouds, or where significant cloud formations occupy different heights within an AIRS pixel. In this initial effort, these scenes often, but not always produce retrievals with poor

spectral fits (described below in Sections 3.7.4 and 3.8) and are filtered out in quality control. We discuss possible ways to improve the forward modelling of clouds in Sec. 6., but we note here that in future versions it may be useful to include an "effective Mie cut-off temperature" as a retrieved parameter in the state vector. Put another way, minimization of the cost function, Eq. 1., would be used to modify the temperature range of the transition from supercooled water to ice clouds. This may more effectively model clouds that contain a mixture of cirrus and supercooled water droplets.

### 35 3.5 Retrieval channels

Table 2 lists the spectral channels used in AIRS-OE retrievals. This channel list is similar to that for AIRS-V6, except only longwave channels were used (< 1650 cm$^{-1}$). We found that using channels in the shortwave region of the AIRS bandpass would often result in retrievals not converging, or producing unrealistic retrieval quantities. This is likely related to errors in calculating



outgoing radiation from reflected sunlight, which remains a challenge in the near-infrared, particularly in cloudy scenes because of uncertainties in the scattering/absorption ratio (Nakajima and King, 1990; Nakajima et al., 1991).

For $O_3$, we do not use the 9.6 μm band as its inclusion often results in a failure for the retrieval to converge. (As noted by Kulawik et al. (2006b) for the TES retrieval, difficulty in finding a global minimum can happen when retrieving all species at

once.) However, including ozone in the retrieval (through its absorption within the 14 μm $CO_2$ band) but not including the strong 9.6 μm band improves the overall fitting, with fewer failed retrievals. Comparisons of $H_2O$ and $T_{atm}$ with validation measurements also improves (not shown). We therefore retrieve $O_3$ only as an "interferent" gas within the 14 μm $CO_2$ region, and avoid the 9.6 μm band.

**3.6 Retrieval by minimization of cost function**

After setting the different elements of the cost function (Eq. 1) as described above, the retrieval is performed by iteratively minimizing the cost function by modifying the retrieval state vector $\hat{z}$ with a combination Gauss-Newton/Levenberg-Marquardt solver. Formulae are described by Bowman et al. (2006), applying the algorithm of Moré (1977). (See also Sarkissian, 2001.) Computational efficiency is improved by retrieving in two steps. First, leaving other variables fixed, only $\tau_{cld}$ is retrieved as this is usually the source of the largest difference between the observed spectra and one calculated from the a priori. The resultant

spectral fit from this initial retrieval can be poor, but this retrieval often provides a better first guess for $\tau_{cld}$ for the second step. In this second step, the $\tau_{cld}$ value is carried over as a first guess (but leaving its a priori value $z_a$ the same) into a simultaneous retrieval of $\tau_{cld}$, $T_{cldtop}$, $r_{eff}$, $T_{sfc}$, $T_{atm}$, $H_2O$, $O_3$ and $CO_2$. Convergence tests are as described in Sect. IV.B of Bowman et al. (2006). If a given retrieval cannot converge within a specified number of iterations, or both the "trust region" and "linearity measure" within the Levenberg-Marquardt solver fall below a specified threshold, the algorithm is halted and flagged as non-convergent.

Converged retrievals are analysed for information content and quality-checked as described below.

**3.7 Information content and error estimation**

**3.7.1 Averaging kernels**

We assume that the retrieval is nearly linear in the vicinity of the solution. The Jacobian is defined as the matrix of derivatives of the outgoing radiance to changes in each element of the state vector,

$$\mathbf{K}_z = \frac{\partial \mathbf{F}(\hat{\mathbf{x}}, \mathbf{b})}{\partial \mathbf{z}} = \frac{\partial \mathbf{F}(\mathbf{M}\hat{\mathbf{z}}, \mathbf{b})}{\partial \mathbf{z}} \qquad (5)$$

and is calculated by finite difference for each retrieval iteration. The gain, $\mathbf{G}_z$, is a measure of the sensitivity of the retrieval, $\hat{\mathbf{z}}$, to changes in the radiance:

$$\mathbf{G}_z = \frac{\partial \hat{\mathbf{z}}}{\partial \mathbf{F}} = (\mathbf{K}_z^T \mathbf{S}_\varepsilon^{-1} \mathbf{K}_z + \mathbf{S}_a^{-1})^{-1} \mathbf{K}_z^T \mathbf{S}_\varepsilon^{-1} \qquad (6)$$

The gain is multiplied by the Jacobian to produce the averaging kernel matrix, $\mathbf{A}$, which is a measure of the sensitivity of the

retrieval vector, $\hat{\mathbf{z}}$, to changes in the true state, $\mathbf{z}$:

$$\mathbf{A} = \frac{\partial \hat{\mathbf{z}}}{\partial \mathbf{z}} = \frac{\partial \hat{\mathbf{z}}}{\partial \mathbf{F}} \frac{\partial \mathbf{F}}{\partial \mathbf{z}} = (\mathbf{K}_z^T \mathbf{S}_\epsilon^{-1} \mathbf{K}_z + \mathbf{S}_a^{-1})^{-1} \mathbf{K}_z^T \mathbf{S}_\epsilon^{-1} \mathbf{K}_z \qquad (7)$$

This is a square matrix dimensioned $n \times n$, where $n$ is the number of elements of the state vector, and as described below, is useful in calculating the error covariance of the retrieval. Each element of the averaging kernel matrix is a measure of the



sensitivity for one retrieved member of a state vector ($\hat{z}_i$) to the changes in the true value of that member ($z_i$), or to the true value of a different member ($z_j$). That is,

$$A_{i,j} = \frac{\partial \hat{z}_i}{\partial z_j} \tag{8}$$

Figure 4 shows a sample averaging kernel from a simultaneous retrieval taken during daytime September 6, 2002 at 26.1°N, 130.6°E, over the Pacific Ocean south of Japan (within the same granule depicted in Fig. 1 of Kahn et al., 2014). A thin cirrus cloud is retrieved with a $T_{cldtop}$ equal to 210.1 K (~155 mb) and a $\tau_{cld}$ of 0.55. The axes indicate the retrieval pressure layers for each constituent, and are not set on a regular altitude or pressure scale. $T_{atm}$, $T_{sfc}$ and $T_{cldtop}$ are retrieved as linear quantities, but $H_2O$, $O_3$, $CO_2$, and $\tau_{cld}$ and $r_{eff}$ are retrieved as their natural logarithms and their partial derivatives are reported as such. Note that

the color scale is restricted to emphasize the weaker sensitivities. The diagonal of the averaging kernel indicates the sensitivity of a constituent to changes of its true self, while (as will be seen) off-diagonals (within the same constituent) have information on the vertical resolution of the retrieval.

A row of the averaging kernel contains measures of the sensitivity of a single element of the state vector to changes in the true values of itself and other elements. Note how in this retrieval, looking across the row for $T_{sfc}$, the retrieved surface temperature

can be sensitive to changes (and errors) in the true surface temperature itself, the lower tropospheric temperature profile, and the cloud properties. There is little to no sensitivity to changes in $H_2O$, $CO_2$, and $O_3$. By contrast, most rows for $CO_2$ indicate retrieval sensitivity to changes in its true self and also changes in all other retrieved constituents.

A column of the averaging kernel contains measures of the sensitivity of the entire retrieval state vector to changes in a single true state vector element. For example, looking up the $T_{sfc}$ column, all constituent retrievals are affected somewhere (even if only

weakly) by sensitivity to changes in the true surface temperature. By contrast, looking at the columns for $CO_2$, changes in the true $CO_2$ tend to affect the retrieved $CO_2$ but not other constituents.

The averaging kernel can be subdivided to provide sensitivity data on scalar values (e.g., $T_{sfc}$ and individual cloud properties), or sensitivity and vertical resolution for individual profiles. Figure 5 shows the retrievals, individual averaging kernels and ancillary information for the $T_{atm}$ and $H_2O$ profiles (using the parts of the averaging kernel matrix that are $\partial \widehat{\mathbf{T}}_{atm}/\partial \mathbf{T}_{atm}$ and $\partial \ln \widehat{[\mathbf{H_2O}]}/$

$\partial \ln [\mathbf{H_2O}]$, respectively.) The leftmost panel shows the a priori and retrieved temperature profile, along with the estimated error (discussed below). For clarity, the error is shown as a separate line (with a separate axis) rather than an error bar. The second panel shows the rows of the $T_{atm}$ averaging kernel, along with its row sums. An averaging kernel can be examined to better understand the sensitivities of the measurement and retrieval. Note, for example, that the temperature averaging kernel rows are comparatively low in the boundary layer ($\gtrsim$ 750 mb), indicating lowered sensitivity to changes in the true temperature, but

sensitivity increases at altitudes above this in the free troposphere. These changes in sensitivity affect the error (in the leftmost panel) which reaches a broad minimum in the region from about 600 to 350 mb. The row sum of an averaging kernel row is a useful indicator of how much a retrieval relies on the spectrum for its results. A row sum near unity indicates that the retrieval at that layer relies mostly from the observed spectral data, while a value near zero indicates mostly reliance on the a priori.

While the row sums are indicative of how much information came from the observation, they do not indicate vertical resolution.

Indeed, a visual inspection of the temperature averaging kernel rows in the left panel of Figure 5 shows that while the row sums are high in the region between about 300 to 100 mb, the widths of the peaks are much broader than those below at higher pressures. To estimate vertical resolution, we use a simple full-width-at-half-maximum calculation for each averaging kernel row, using a Gaussian fit and assuming a 7 km scale height in converting pressure to altitude. (Other approaches are described in Ch. 3.3 of Rodgers, 2000.) From this fitting approach, shown in the third panel of Figure 5, the vertical resolution is about 1 to





1.5 km from the ground to about 300 mb, above which the resolution of the $T_{atm}$ retrieval quickly degrades. The fitting approach used here is not useful at pressures less than about 200 mb where the rows of the $T_{atm}$ averaging kernel become much flatter, and can be double-peaked when crossing the tropopause.

The right three panels of Figure 5 show the $H_2O$ a priori, retrieval and error, averaging kernel with row sums, and estimated

vertical resolution. (As the logarithm of the $H_2O$ volume mixing ratio is retrieved, the upper and lower errors of the retrieval in linear space are slightly different. They are shown as separate lines rather than error bars for clarity.) As with $T_{atm}$, the averaging kernel at pressures greater than 750 mb indicates lower sensitivity in the boundary layer. Sensitivity improves and is fairly constant between 750 mb up to about 200 mb, above which sensitivity decreases and effectively disappears at 100 mb. The rightmost panel of Figure 5 shows the approximate vertical resolution. The "flattened" averaging kernels near the boundary layer

leads to a local maximum in the vertical resolution of ~3 km is seen at about 800 mb, but the averaging kernel rows become more sharply defined at lower pressures. This improved definition is reflected in the vertical resolution, which is about 1.4 km at 700 mb, and steadily increases above to a maximum of 4.6 km at 200 mb.

Note for an AIRS retrieval, an averaging kernel is scene dependent. Sensitivities at different layers depend on the amounts of trace gases present, the temperature lapse rate, the particulars of the cloud field, the view angle, and the spectral channels

employed. Since the AIRS-OE retrievals are simultaneous and not sequential, the averaging kernel describes dependencies within and between retrievals of different constituents, and can be used to more robustly calculate uncertainties as described below.

### 3.7.2 Error estimation

The smoothing error covariance measures the uncertainty in the fine structure of the retrieval due to the measurement's limited

vertical resolution. However, as we have an averaging kernel from a joint retrieval, the smoothing error also indicates how the uncertainty in one retrieved constituent affects the uncertainty in another:

$$\mathbf{S}_s = (\mathbf{A} - \mathbf{I}_n)\mathbf{S}_a(\mathbf{A} - \mathbf{I}_n)^T \tag{9}$$

(See Sect. V(B) of Bowman et al., 2006, and Sections 3.4 and 4.1 of Rodgers, 2000.)

The retrieval noise error covariance calculates the impact of the radiance noise on the retrieval:

$$\mathbf{S}_m = \mathbf{G}_z \mathbf{S}_\varepsilon \mathbf{G}_z^T \tag{10}$$

With substitutions, these terms can be added to provide the covariance of the maximum a posteriori solution:

$$\hat{\mathbf{S}} = (\mathbf{K}_z^T \mathbf{S}_\varepsilon^{-1} \mathbf{K}_z + \mathbf{S}_a^{-1})^{-1} \tag{11}$$

with the square roots of the diagonal reported as errors for the state vector. Note also that the total retrieval error does not include any errors systematic as input to, or from the forward model (e.g., instrumental lineshape errors, spectral biases or other errors

that are correlated across observations.) We again emphasize that since our a priori covariances are ad hoc, caution should be observed in using the reported errors.

For this initial version of our algorithm, we have not implemented code to calculate the model parameter error, which contains the uncertainty from parameters affecting the retrieval, but are not retrieved themselves (e.g., surface pressure, emissivity, scan angle, etc.):

$$\mathbf{S}_{mp} = \mathbf{G}_z \mathbf{K}_{Psurf} \mathbf{S}_{a,Psurf} (\mathbf{G}_z \mathbf{K}_{Psurf})^T + \mathbf{G}_z \mathbf{K}_{emis} \mathbf{S}_{a,emis} (\mathbf{G}_z \mathbf{K}_{emis})^T + \cdots \tag{12}$$

In this case, the total retrieval error covariance would be the sum of Eqs. (11) and (12):



$$\mathbf{S}_{\text{tot}} = \hat{\mathbf{S}} + \mathbf{S}_{\text{mp}} \tag{13}$$

The addition of the model parameter error (Eq. 12) is planned for future development.

For constituents retrieved in logarithmic space, the error reported for the i'th element, $\epsilon_i$, is the error in the logarithm of the retrieved value, $\hat{z}_i$, with the range [lower, upper] of the retrieval in linear space being:

$$[\exp(\hat{z}_i - \epsilon_i), \exp(\hat{z}_i + \epsilon_i)] \tag{14}$$

### 3.7.3 Calculation of Relative Humidity and Error

In calculating relative humidity (RH), we use the layer retrievals of temperature and water vapor. Eqs. (2.5) and (2.21) of Wagner and Pruß (2002) are used to determine saturation pressures of water vapor over liquid and ice. At temperatures between 253.15 and 273.15K, we set saturation pressure as a sliding-scale weighted average of those over ice and over water. The relative

humidity error calculation uses recalculated RHs adding the errors from the temperature and (separately) the positive, linear value of the water vapor error (the right-hand side of Eq. 14). We report the relative humidity uncertainty as the root-sum-of-squares of the differences between these re-calculated relative humidities and the reported values.

### 3.7.4 Chi Square Fitting Parameter

The chi square fitting parameter, $\chi^2$, is a goodness-of-fit statistic of how well a spectrum's radiance is fitted within the bounds of

the radiance error:

$$\chi^2 = \frac{1}{N} \sum_{i=1}^{N} \left( \frac{y_i - [\mathbf{F}(\mathbf{x}, \mathbf{b})]_i}{\varepsilon_i} \right)^2 \tag{15}$$

where N is the number of channels, and $\varepsilon_i$ is the radiance error in channel $i$. A $\chi^2 \gg 1$ indicates a poor spectral fit to the observed radiance. While the $\chi^2$ does not directly enter into the error characterization, it is used in quality control as described below.

**3.8 Quality control filtering**

Retrievals that do not meet the following three criteria are filtered out:

1.   Normal convergence within the maximum specified number of iterations,

2.   Chi square fitting parameter, $\chi^2 < 3$, and

3.   Retrievals in layers with $T_{\text{atm}} > (T_{\text{cldtop}} - 10 \text{ K})$ require a surface temperature averaging kernel $> 0.6$.

The first criterion is to avoid waste of computational resources on poorly- or non-converging retrievals. The second criterion is to avoid reporting profiles with poor spectral fits. This often happens under ice cloud conditions when the cloud optical depth is high ($\gtrsim 20$); it's likely that the radiative transfer is incorrectly calculated because a cloud is assumed to "fit" in one vertical model layer while in reality, thick clouds extend over many model layers. Poor spectral fits can also often occur when there was a high standard deviation, $\gtrsim 20\text{K}$, of the MODIS 1 km cloud-top temperature weighted over the AIRS spatial response function.

Again, we suspect that this poor fitting is from limitations in our forward model which is limited to one cloud layer; the radiative transfer calculation can be inadequate when there were several cloud tops at different temperatures within the AIRS footprint.

The third criterion is a means to remove layers of a profile that have unphysical values of the relative humidity calculated from the retrieved temperature and water vapor; these are usually in or near the boundary layer. We again note that a cloud's transmissivity is incorporated in only one layer of the forward model vertical pressure grid, no matter how thick the cloud. We





hypothesize that this can lead to erroneous outgoing radiances for temperature and water vapor channels in regions at or below moderately thick clouds, which in turn, produces erroneous Jacobians and averaging kernels. However, surface temperature retrievals appear to more correctly give a low-to-zero averaging kernel under moderate-to-thick cloud optical depths. We therefore require that retrievals at layers below clouds must "see" the surface (determined by the $T_{sfc}$ averaging kernel having a

minimum of 0.6). For retrievals above clouds where the surface temperature averaging kernel is less than 0.6, an additional thermal contrast provided by 10 K buffer between the cloud top and the lowest profile layer to pass quality control eliminates more unphysical retrievals. Most, but not all of the retrievals that produce unphysically high relative humidities are eliminated by this method, usually in the boundary layer.

## 4 Results

For initial examination of cloud, temperature and $H_2O$ profile retrievals for this effort, we use results of an AIRS daytime granule (#44) over the subtropical western Pacific Ocean from September 6, 2002. This is the same granule examined and discussed in detail by Kahn et al. (2014) for AIRS-V6 cloud products. This granule has a large mix of cloud types and weather regimes, including a tropical cyclone to the west, and a mix of low and mid-level clouds from the center to the east and to the south. In the following figures, to provide a large-scale meterological context, MODIS-derived data are also shown including those footprints

where the AIRS-OE retrieval failed. (For an AIRS granule, there are 90 observations on the cross-track, and 135 observations along-track. Here, the yield of retrievals passing quality control for at least some layers is ~57% of 12150 observations.) In the following sub-sections, we compare our results to those of different algorithms operating on the same scenes. "MODIS-avg" retrievals are from the MODIS 1 km pixels (MYD06 dataset) in a weighted average over the AIRS SpatialRF, as described above in Sect. 3.2.2.

**4.1 Cloud-top temperature, effective cloud optical depth, and effective cloud particle radius**

Figure 6 displays cloud-top temperatures ($T_{cldtop}$), effective optical depths ($\tau_{eff}$), and effective particle radii ($r_{eff}$) from the a priori (left column) and AIRS-OE retrievals (center column) from the granule described above, along with retrieval averaging kernels (right column). With few exceptions, the a priori are generated from co-located MODIS-avg retrievals. The morphology of the AIRS-OE retrieval fields are similar to the a priori, and the morphology of the averaging kernel fields are similar to each other.

Sensitivity is enhanced for ice clouds because of the higher thermal contrast with the surface. For all clouds as they get thicker, their infrared radiation is more dominant in the window channels, producing a more confident retrieval. An examination (not shown) indicates that the $T_{cldtop}$ averaging kernels reach ~0.5 at cloud optical depths of about 0.4 for ice clouds, and between 1 and 2 for water clouds. (An averaging kernel of 0.5 indicates that roughly half the information of the retrieval is from the spectral data and half is from the a priori.) We do note, however, that the retrievals can fail quality control at or near the thick center of

the cyclone, likely for reasons described in Sect. 3.4.

**4.2 Comparison with CloudSat/CALIPSO**

We make an additional "snapshot" comparison with (multilayer) cloud observations by the combined CloudSat cloud profiling radar (Stephens et al., 2002) and the Cloud-Aerosol Lidar and Infrared Pathfinder Satllite Observations lidar (CALIPSO; Winker et al., 2007), using the 2B-CLDCLASS-LIDAR product (Wang et al., 2013). Figure 7 compares AIRS-OE cloud-top retrievals

against near-coincident CloudSat/CALIPSO (CsC) observations, using the same date and region as in Figure 2 of Wang et al. (2016), who compared MODIS cloud classifications and CsC profiles. (As did Wang et al. (2016), the horizontal axis is ordered





by decreasing latitude.) The upper two panels show AIRS-OE cloud optical depths and cloud-top temperatures from a daytime July 31, 2009 swath over the Pacific (latitude on the horizontal axis) with the CsC transect overlaid. The third panel shows the vertical extent of the clouds from CsC (in grey). Superimposed are the approximate cloud-top altitudes of the QA-passed AIRS observations (no more than one per AIRS cross-track) closest to the CsC transect. (Note that the distance between the center of

an AIRS observation and the closest CsC observation can be up to 7.5 km). The AIRS observations in this panel are colored by the AIRS-OE retrieved cloud optical depth. The bottom panel again shows the CsC cloud layer and AIRS-OE cloud-top altitudes, but colored by the AIRS-OE cloud-top temperature averaging kernel, which can be used as a measure of confidence in the AIRS cloud-top altitude.

Similar to the (1 km footprint) MODIS retrieval on the same transect (see Figure 2 of Wang et al., 2016), AIRS-OE retrieves

cloud tops at a lower altitude than CsC for the thicker regions of the cirrus clouds (marked "A" and "B" in Figure 7). This is similar to comparisons of AIRS Version 5 cloud retrievals to CsC by Kahn et al. (2008), citing Holz et al. (2006) in how infrared retrievals of cirrus tend to place the cloud-top 1 to 2 km or more below the physical cloud top. The retrieval does not pass quality control at all points over the deep convective core, likely because radiance spectra above such deep clouds can be poorly calculated by our forward model. Significant differences in cloud-top altitude are notable along the thin cirrus between about 5°

and 12° ("C"), with retrieved optical depths of about unity or less. However, a close examination of the cloud-top temperature and cloud optical depth fields in the upper panels between 10°N and 5°N show that the CloudSat/CALIPSO transect is on the edge of a thin north-south aligned strip of ice cloud, and the center of the closest AIRS observations to the CsC transect were between 2 to 5 km apart, so some sampling bias may be present. Similarities and differences shown in Figure 7 may be broadly similar to previous AIRS/MODIS/CloudSat/CALIPSO comparisons (e.g., Kahn et al., 2007; 2008), but here it illustrates the

feasibility of forward-modeling clouds explicitly in a hyperspectral IR retrieval, simultaneous with temperature and trace gases.

### 4.3 Temperature, water vapor and relative humidity profiles

Figure 8 presents maps of $T_{atm}$ (top row), $H_2O$ (middle row) and a calculated relative humidity (RH; bottom row) at the 918 mb layer for the September 6, 2002 granule discussed above. For comparison, the left column shows results from the operational AIRS-V6 retrievals, interpolated by log(pressure) to the AIRS-OE retrieval layer.  Note that the AIRS-V6 retrievals used cloud-

cleared radiances on the AMSU footprint, each point covering an area approximately nine times that of a single AIRS infrared footprint. The second column shows the initial guess for each of these, calculated by linear interpolation in time and space and vertically by log(pressure) from 6 hour ECMWF analyses. The third column shows the AIRS-OE retrievals passing the quality control criteria described in Sec 3.7. The fourth column shows the estimated error of the AIRS-OE retrieval. Note that $H_2O$ is presented in volume mixing ratio, not mass mixing ratio. Comparing AIRS-OE relative humidity to its a priori or AIRS-V6

shows significant local differences in relative humidity, and many unphysically high values (> 100%) throughout the region studied. We note, however, that the calculated error for the AIRS-V6 relative humidity at this layer is fairly high, with a median of 28.7% with an interquartile range (IQR, that is, the range between the 25[th] and 75[th] percentiles) of 26.0 to 31.9%. The AIRS-OE relative humidity is biased high compared to AIRS-V6, with a median bias of 9.0% (IGR = -1.1 to 19.7%).

At 525mb (Figure 9), qualitative agreement for $T_{atm}$, $H_2O$, and RH is improved compared to 918 mb across the ECMWF-derived

a priori, AIRS-V6 and AIRS-OE. However, the AIRS-OE retrieval for $H_2O$ begins to depart from the a priori to more closely resemble the AIRS-V6 retrieval. For example, the a priori shows a "curl" of higher water vapor through drier air near 30°N and 135°E (blue color overlaid by "A") while AIRS-V6 and AIRS-OE show this region to be more uniform. The AIRS-OE median relative humidity error is 16.1% (IQR = 12.0 to 19.6%), smaller than those at the 918 mb layer. Regional biases in the $H_2O$ and





RH of AIRS-OE compared to AIRS-V6 can again be readily seen, although the median RH bias is reduced to -0.5 % (IQR = -4.3 to 6.4%).

At 321 mb (Figure 10), the AIRS-OE temperature field shows a much broader region of cold air than either AIRS-V6 or the a priori (dark blue overlaid by "A"). However, the $H_2O$ and RH fields of AIRS-V6 and AIRS-OE more resemble each other than

the a priori, and the additional horizontal resolution of AIRS-OE allows sharper boundaries to be seen between dry and wet regions. Although there are still many missing pixels in AIRS-OE, there is improved definition in the boundaries between lower and higher values of the $H_2O$ volume mixing ratio ("B" in upper left). For relative humidity, note that the region of humid air near 32°N and 140°E in AIRS-V6 ("C") is more fully resolved as three small but distinct regions in AIRS-OE. The median error of the AIRS-OE relative humidity is 13.2% (IQR = 7.2 to 19.6%). The median bias of the AIRS-OE relative humidity compared

to AIRS-V6 is 0.2% (IQR = –4.0 to 6.9%), similar to the 525 mb layer.

As a test of the algorithm's sensitivity to the infrared spectrum, Figure 11 compares the 321 mb retrieval of relative humidity under different a priori. The left panels show the relative humidites calculated from $T_{atm}$ and $H_2O$ a priori as interpolated from ECMWF analyses (as above), a climatology, and the neural-net a priori of the current AIRS-V6 retrieval scheme. The right panels show the AIRS-OE relative humidity retrievals from these different a priori. In all cases, MODIS-avg data are used as

cloud a priori. While showing some differences in the details, all retrieval relative humidity fields show a similar structure, indicating a robust sensitivity of the retrieval in the free troposphere over the ocean.

## 5 Comparison with radiosondes

### 5.1 MAGIC campaign sondes

AIRS-OE retrievals of temperature and water vapor are compared with radiosonde profiles from the ship-based Marine ARM

GPCI Investigation of Clouds (MAGIC) campaign of September, 2012 through October, 2013. Sondes were launched from a Department of Energy Atmospheric Radiation Measurement mobile facility atop a container ship travelling between Honolulu and Los Angeles. An extensive description of the MAGIC field campaign, and comparison of AIRS-V6 and ECMWF ERA-Interim reanalysis water and temperature results with MAGIC has been reported by Kalmus et al. (2015). Figure 12 illustrates the location of the sondes matched to AIRS (launched within 3 hours and 100 km of an AIRS observation), colored by the

number of successful retrievals at 321 mb; anywhere from 3 to 105 successful AIRS-OE observations are achieved as matchup to a single sonde.

Whether an AIRS-OE retrieval is successful at a particular pressure layer depends in large extent on the particularities of the cloud cover, and so the number of successful retrievals can vary widely for matched radiosondes. For example, of the 210 MAGIC sondes that were coincident with AIRS observations, an average of $46 \pm 22$ (1σ) AIRS-OE successful retrievals were

made for each sonde at 321 mb, while lower down at 918 mb, an average $30 \pm 21$ retrievals were made (not shown). Simply taking a global average (or even a global median) of the differences between AIRS retrievals and sonde observations can introduce significant sampling biases, with clear or nearly clear areas over-represented. We also found that average bias could often be significantly skewed from unphysical retrieval outliers – usually because the water vapor retrieval was too high. Thus, to calculate an overall bias between AIRS-OE and the MAGIC sondes, and compare this to the a priori and AIRS-V6 results, we

report the "median of the medians," calculated in this manner:

(a) For the ensemble of a priori and QA-passed retrievals for a single sonde, we calculate the median $T_{atm}$ bias, RH bias, and relative bias in $H_2O$ (e.g., [AIRS – sonde] / sonde) in %).





(b) We do the the same as above for AIRS-V6 observations, but only where a successful AIRS-OE retrieval is within an AIRS-V6 footprint, and only layers that have an AIRS-V6 quality control of 0 (best) or 1 (good). As there can be up to nine AIRS-OE retrievals within an AIRS-V6 footprint, an AIRS-V6 retrieval is only entered once in calculating the median to avoid over-counting.

Note that these criteria for matching up sondes and AIRS-V6 observations are different from Kalmus et al. (2015), so caution should be taken in comparing results here with that work. Also, since the inclusion of an AIRS-V6 profile in calculations depends on the QA result of an AIRS-OE retrieval, these data should not be taken as validation of AIRS-V6 results. Note also that the AIRS-V6 uses a different a priori than AIRS-OE, and the AIRS-V6 a priori is not shown or compared here.

    Figure 13 illustrates the median temperature profile differences between the a priori, AIRS-OE, and AIRS-V6 retrievals for $T_{atm}$,
the relative difference for $H_2O$, and the difference in RH compared to the MAGIC sondes. Thin lines indicate the 25th and 75th percentiles of the distributions. For temperature, AIRS-OE (blue line) shows a negative bias of 0.65K at the surface, and increasing to a maximum of 0.9K at 865mb. The positive bias continues to about 400 mb, and is higher than either the a priori or AIRS-V6. Between 400 mb and 200 mb, the AIRS-OE retrieval is within 0.3K, as are the a priori and AIRS-V6. For water vapor, the (global median) relative bias of AIRS-OE retrievals stays within 10% of the sondes from the surface to about 800 mb,
where it increases to about a 20% bias at 525 mb, decreases to a 3% bias at 321 mb, and then increases again, as do the a priori and AIRS-V6 retrievals. For the bias in relative humidity, except for a local minimum of -6% at 865 mb, biases are positive and tend to be within 5% up to 200mb, but this good agreement may be partly due to compensating biases in temperature and water vapor.

## 5.2 Tropical Western Pacific (TWP),  Southern Great Plains (SGP)  and North Slope of Alaska (NSA) sondes

For these comparisons, we use AIRS observations co-located with high quality radiosondes launched from ground-based sites of the U. S. Department of Energy's Atmospheric Radiation Measurement (ARM) Climate Research Facilities. Inclusion criteria and median bias calculations are similar to those for the MAGIC sondes (Sec. 5.1).  Figure 14 illustrates the median temperature biases, water vapor relative biases and relative humidity biases for the ECMWF analysis-derived a priori, and the AIRS-OE and AIRS-V6 retrievals over the TWP site at Nauru, the SGP site in Oklahoma, and the NSA site in Alaska.

For temperature, all three sites show AIRS-OE median biases for temperature below ±1 K, and median relative biases for $H_2O$ to mostly within ±10% . At TWP, AIRS-OE generally shows larger biases compared to the a priori for temperature, but does as well or better than the ECMWF-derived a priori for $H_2O$ and relative humidity from about 800 mb up to about 200 mb. At SGP, AIRS-OE shows a of high bias of ~0.7 K between the surface to 400 mb, but slightly improves upon the a priori water vapor for most of the troposphere, except near 700mb. AIRS-OE relative humidity biases compare well with the a priori. At NSA, AIRS-
OE temperature biases are low at altitudes above ~800 mb, as are water vapor relative biases to about 200 mb. Relative humidities are improved compared to the a priori from the surface to about 400 mb.

    While again we caution that the results in Figure 14 should not be taken as validation for AIRS-V6, we note that the single footprint retrieval AIRS-OE results compare well with the cloud-cleared spectra results of AIRS-V6. In particular, we note the improvement in relative humidity retrievals throughout most of the troposphere at TWP, and in the boundary layer for SGP and
NSA.





## 6 Summary and Discussion

We have presented a new retrieval scheme for the AIRS instrument, AIRS-OE, which uses MODIS Level 2 data as cloud parameter a priori, and a forward model that incorporates cloud effects in its radiative transfer. As AIRS-OE directly uses L1b spectra in retrievals, and not L2 cloud-cleared spectra as done in AIRS-V6, it improves the nadir horizontal resolution over

AIRS-V6 from ~45 km to ~13.5 km. Focusing on cloud parameters, and temperature and water vapor profiles, we have presented some initial comparisons to currently operational AIRS Version 6 products. The improved horizontal resolution has been shown to provide greater detail in water vapor and relative humidity fields in the free troposphere. Initial tests indicated a robust retrieval sensitivity using different a priori. As AIRS-OE rests on an optimal estimation framework, and includes simultaneous retrieval of profiles and scalar variables, it has an information content analysis that operates both within and across different

atmospheric parameters. Initial comparisons against co-incident radiosondes indicate that retrieval biases for temperature and water vapor profiles are at least competitive with AIRS-V6.

Some aspects of this new retrieval need development. More realistic clouds in the forward model, with multiple cloud decks or clouds that extend over several model layers, will likely allow more footprints to be successfully analysed. A better a priori $\tau_{cld}$ at nighttime may be made by comparing brightness temperatures in the window channels to the a priori surface temperature, similar

to Kulawik et al. (2006a). Improved cloud results will hopefully better leverage and compliment the spatial coverage and horizontal resolution of MODIS, and the vertical precision and detail of CloudSat/CALIPSO. A forward model incorporating scattering by dust and other aerosols would open more regions for analyses. Efforts can be made in better modeling the outgoing daytime radiance of the shortwave channels of AIRS (> 2200 cm$^{-1}$), which now limit us in fully exploiting the spectral range of the instrument, particularly near the 4 μm $CO_2$ band. An observation-based a priori covariance (including cross-species

covariances), specific to region and season, would provide improved constraints and more realistic retrieval errors. A method of assigning ground-level snow and ice conditions to observations is needed – possibly from the co-located AMSU microwave instrument or other near-real-time data (see Pope et al., 2014). Addition of effective emissivity retrievals will likely be necessary to improve results over "difficult" regions, such as deserts or mountains, and could include some weighting by, say, the MODIS cloud mask to determine an a priori emissivity as it affects the AIRS observation by taking cloud cover into account. The

algorithm could be extended to better retrieve $O_3$ using its 9.6 μm band, as well as retrieve $CH_4$, CO and other gases within the AIRS bandpass. Successful implementation of these improvements may be challenging (or at least time-consuming) but could prove useful not just to single-footprint retrieval from AIRS, but to other instruments such as CrIS and IASI.

Since the design stage of the AIRS instrument in the 1990's, increased computing power and advances in modeling cloudy spectra allow new approaches to utilize the high spectral resolution output of existing infrared sounders. The horizontal

resolution gained by avoiding cloud-clearing can provide more nuanced data for water vapor, especially where it is highly variable at smaller spatial scales. Even with the algorithmic liens described, results presented here indicate AIRS-OE retrievals on cloudy infrared spectra can compare well with operational AIRS-V6 retrievals that require cloud-clearing. Improvements in execution speed and data handling may be needed before this work can become operational like AIRS-V6. However, with some 16 billion AIRS infrared spectra since launch in 2002, numerous opportunities are available for targeted studies with this new

algorithm.

## Acknowledgments

We thank Kevin Bowman, Sergio DeSouza-Machado, John Gieselman, Michael Gunson, Glynn Hulley, Susan Kulawik, Bjorn Lambrigsten, Evan Manning, Tom Pagano, Edwin Sarkissian, L. Larrabee Strow, Joao Teixeira, Geoff Toon, Tao Wang, and



John Worden. This research was primarily funded through NASA's "The Science of Terra and Aqua" grant program, and performed at the Jet Propulsion Laboratory, California Institute of Technology, under contract to the National Aeronautics and Space Administration. Copyright 2017 California Institute of Technology. U. S. Government sponsorship acknowledged.

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





| Constituent | Covariance along diagonal | Covariance off diagonal length scale (km) |
|---|---|---|
| Temperature profile | $(2K)^2$ | 0.5 |
| $H_2O$ | $[\log(1.4)]^2$ from ground to 100mb then gradually reduced to $[\log(1.01)]^2$ at 50 mb and above | 0.5 |
| Surface temperature | $(2K)^2$ | N/A |
| $O_3$ | $[\log(1.1)]^2$ from ground to 100 mb then gradually increasing to $[\log(1.2)]^2$ at 50 mb and above. | 3.0 |
| $CO_2$ | $[\log(1.02)]^2$ | 3.5 |
| Cloud optical depth (t) | $[\log(2.)]^2$ . | N/A |
| Cloud-top temperature | $(4K)^2$ if cloud-top temperature can be calculated from MODIS<br><br>$(25K)^2$ if cloud-top temperature cannot be calculated from MODIS, but MODIS data indicate thin cirrus | N/A |
| Cloud particle size | $[\log(2.)]^2$ | N/A |

Table 1: A priori covariances used for retrievals.





| Table 2: Retrieval Channels (cm$^{-1}$) | | | | | | | |
|---|---|---|---|---|---|---|---|
| 662.02 | 678.57 | 700.77 | 718.87 | 752.09 | 820.84 | 1384.47 | 1513.83 |
| 664.51 | 681.46 | 701.05 | 719.17 | 753.38 | 839.92 | 1389.39 | 1519.07 |
| 666.26 | 681.72 | 702.74 | 719.46 | 753.70 | 847.14 | 1392.15 | 1521.05 |
| 666.77 | 689.49 | 703.87 | 719.76 | 755.00 | 849.57 | 1397.13 | 1524.35 |
| 667.27 | 689.76 | 704.43 | 720.94 | 755.32 | 880.40 | 1407.77 | 1541.77 |
| 667.78 | 691.12 | 706.13 | 721.54 | 758.26 | 917.30 | 1419.15 | 1544.48 |
| 668.28 | 691.39 | 706.99 | 721.83 | 768.88 | 937.90 | 1427.22 | 1547.20 |
| 668.54 | 692.75 | 707.84 | 723.03 | 769.89 | 948.18 | 1432.47 | 1551.30 |
| 668.79 | 693.02 | 708.70 | 723.32 | 773.28 | 979.13 | 1436.57 | 1554.04 |
| 669.04 | 694.12 | 709.56 | 724.52 | 776.36 | 1121.00 | 1441.88 | 1556.10 |
| 669.55 | 694.40 | 711.00 | 726.32 | 778.08 | 1134.46 | 1462.09 | 1560.24 |
| 669.81 | 694.67 | 711.29 | 732.61 | 779.11 | 1218.49 | 1468.82 | 1563.02 |
| 670.06 | 695.77 | 712.74 | 734.15 | 790.33 | 1225.13 | 1471.91 | 1572.09 |
| 670.57 | 696.05 | 714.19 | 738.48 | 793.89 | 1310.18 | 1474.38 | 1586.26 |
| 672.10 | 697.43 | 714.48 | 740.03 | 795.68 | 1315.47 | 1479.36 | 1598.50 |
| 673.64 | 697.71 | 715.94 | 742.85 | 798.92 | 1340.19 | 1483.74 | 1605.05 |
| 675.19 | 698.82 | 717.40 | 746.01 | 801.10 | 1367.25 | 1493.21 | |
| 676.75 | 699.10 | 717.99 | 747.60 | 803.65 | 1376.88 | 1498.96 | |
| 677.01 | 699.38 | 718.28 | 749.20 | 804.75 | 1379.58 | 1500.88 | |
| 678.31 | 699.66 | 718.58 | 750.48 | 811.79 | 1381.21 | 1502.16 | |

**Table 2: AIRS retrieval channel frequencies used for this study.**





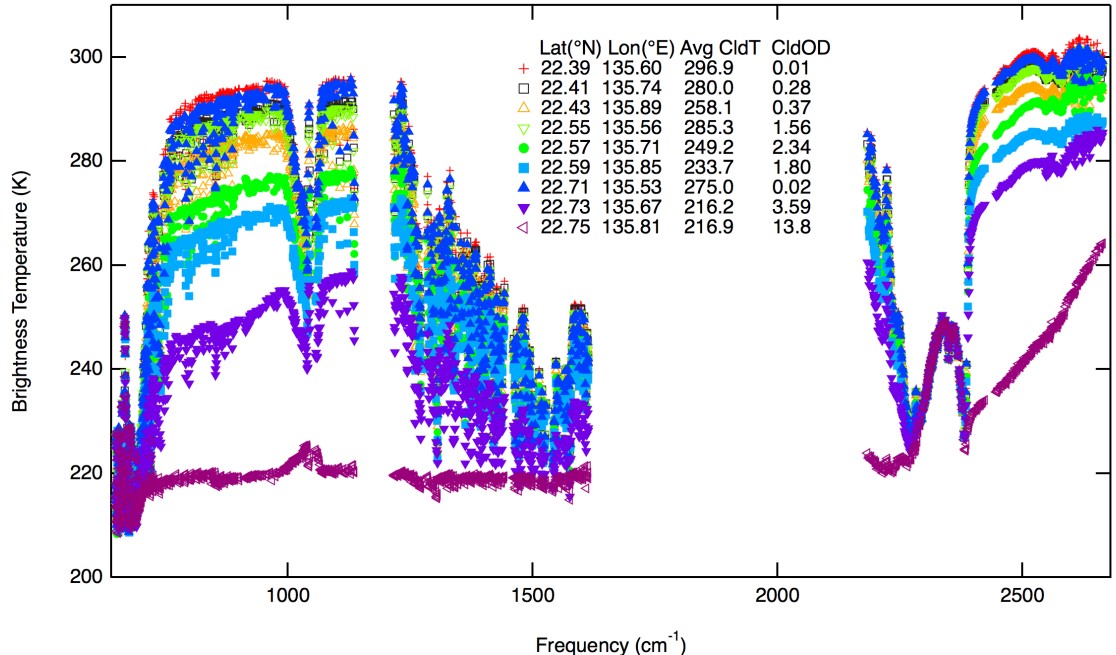

**Figure 1:** AIRS Level 1b brightness temperature observations of adjacent cloudy spectra. Data are from daytime Granule 44, Sept. 6, 2002. Average cloud-top temperatures and cloud optical depths are estimated from coincident MODIS L2 retrievals, averaged on the AIRS spatial response function (see Sect. 3.2.2 in text.)





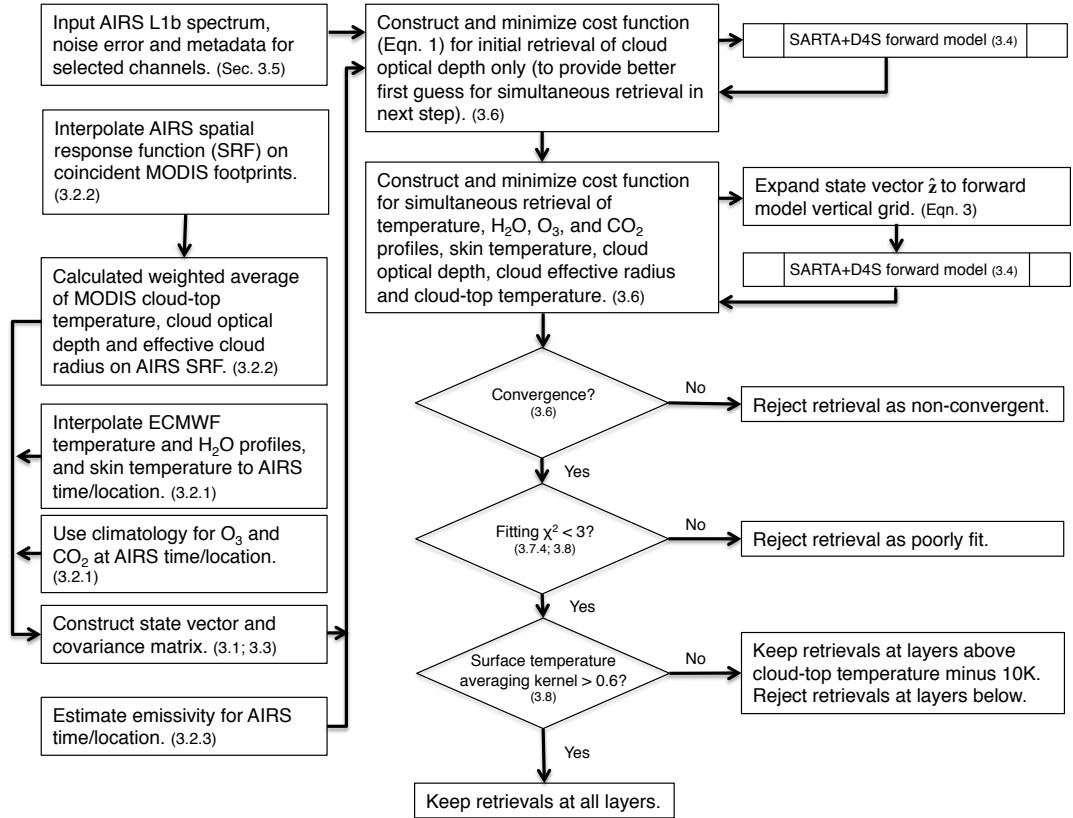

**Figure 2:** Simplified block diagram of AIRS-OE retrieval algorithm for cloudy scenes. Blocks are annotated with section numbers from this paper for further information.





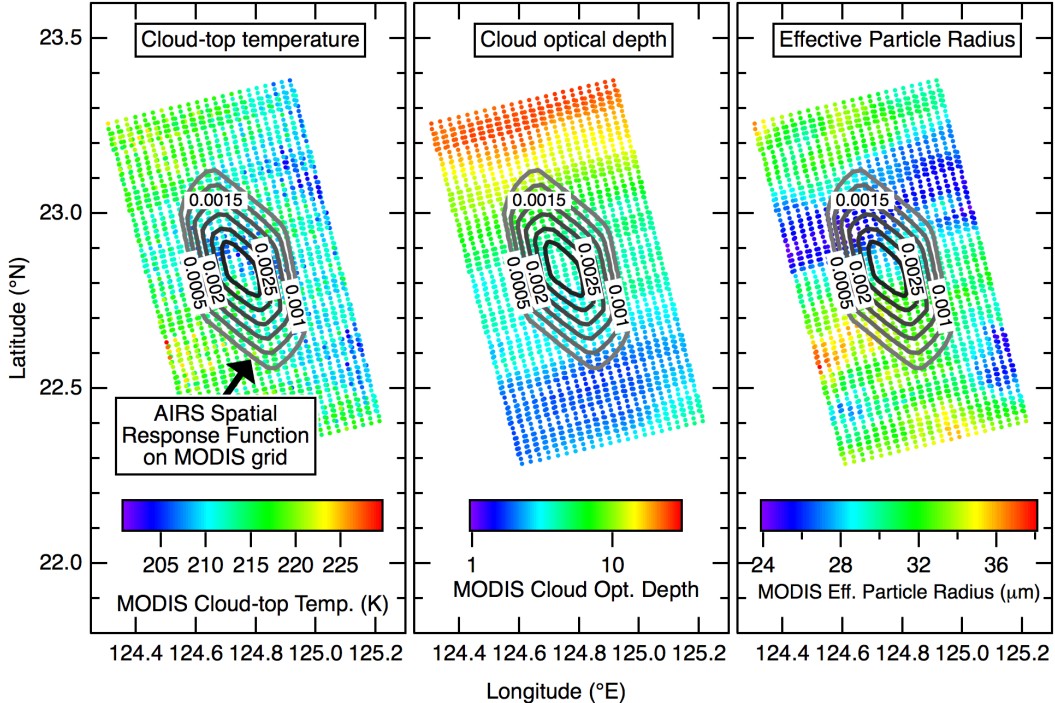

**Figure 3:** Sample MODIS fields of cloud-top temperature, cloud optical depth and effective particle size overlaid with the AIRS Spatial Response Function (SpatialRF) interpolated to the MODIS grid. Data are daytime observations over ocean from September 6, 2002. For this example, the weighted averages over the AIRS SpatialRF are 211.3 K for cloud-top temperature, 4.26 for optical depth, and 29.5 µm for effective particle radius.





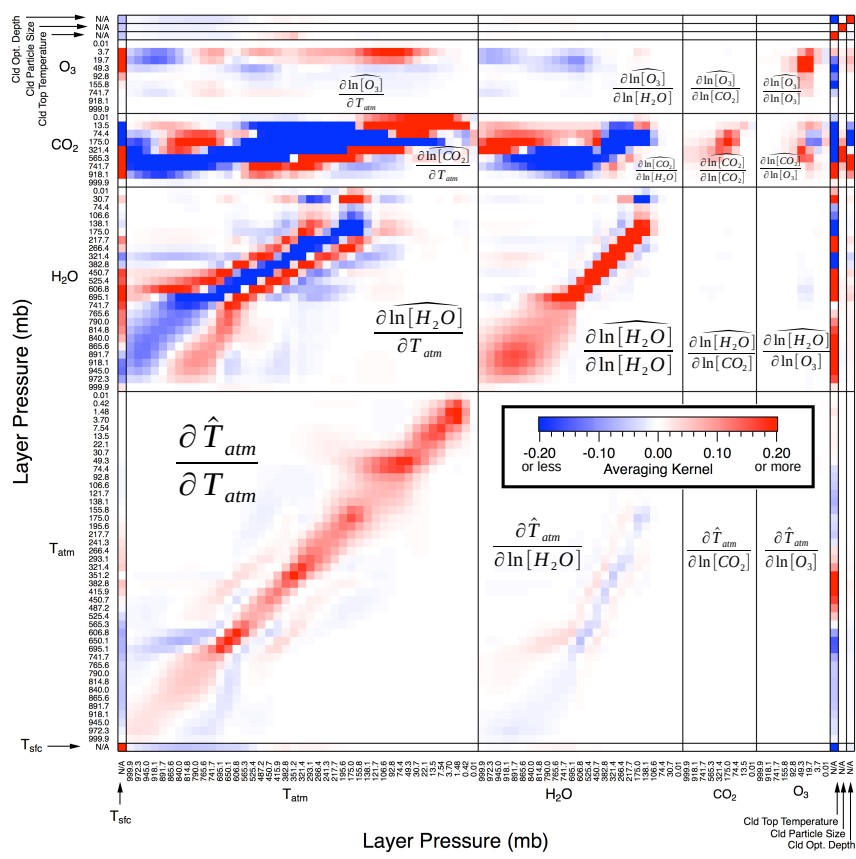

**Figure 4:** Sample averaging kernel from simultaneous retrieval. A row of the averaging kernel matrix is a measure of the sensitivity of the retrieved value to changes in the "true" value of itself and other parameters, shown in the columns, assuming the retrieval is in a near- linear regime. A column indicates the sensitivity of the retrieved state vector to a change in the true value of a single retrieval parameter. The color scale has been limited to better show the weaker sensitivities.



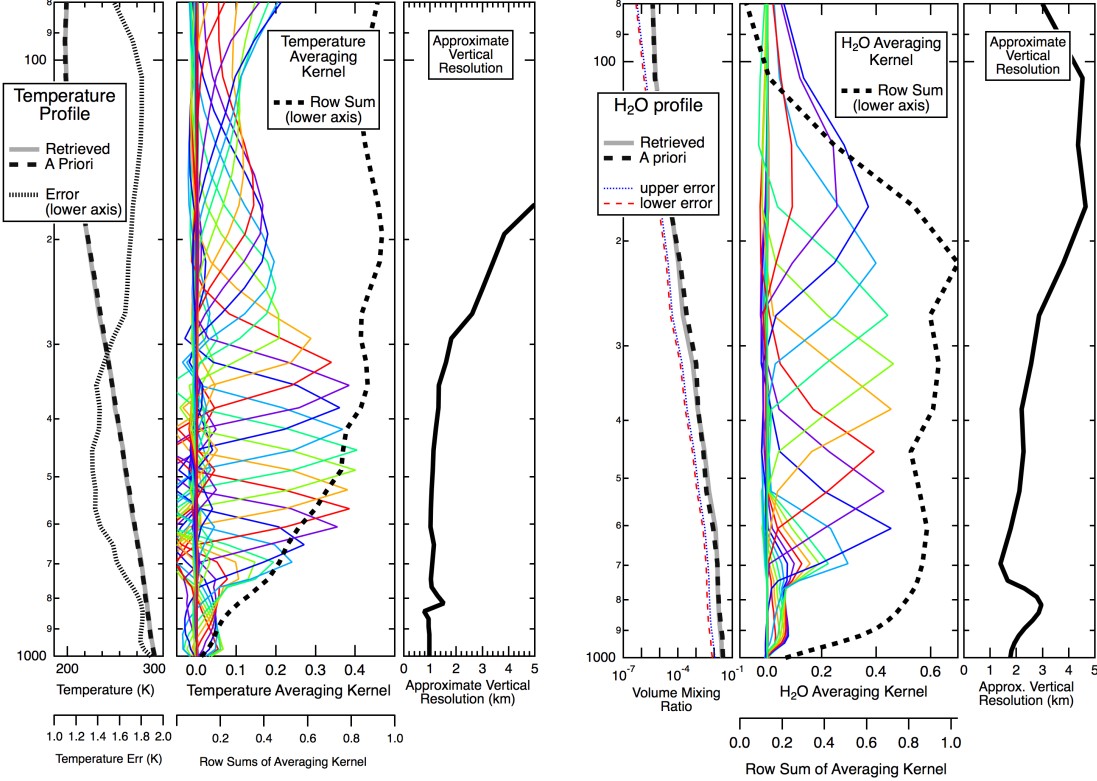

**Figure 5:** Sample retrieval profiles, errors, averaging kernels, row sums of the averaging kernels, and approximate vertical resolutions for temperature (left panels) and water vapor volume mixing ratio (right panels). Each colored line in the averaging kernel panels is from a partial row of the averaging kernel (e.g., the rows of $\partial\widehat{\mathbf{T}}_{atm}/\partial\mathbf{T}_{atm}$ and $\partial\ln[\widehat{\mathbf{H_2O}}]/\partial\ln[\mathbf{H_2O}]$, as seen in Fig. 4.)





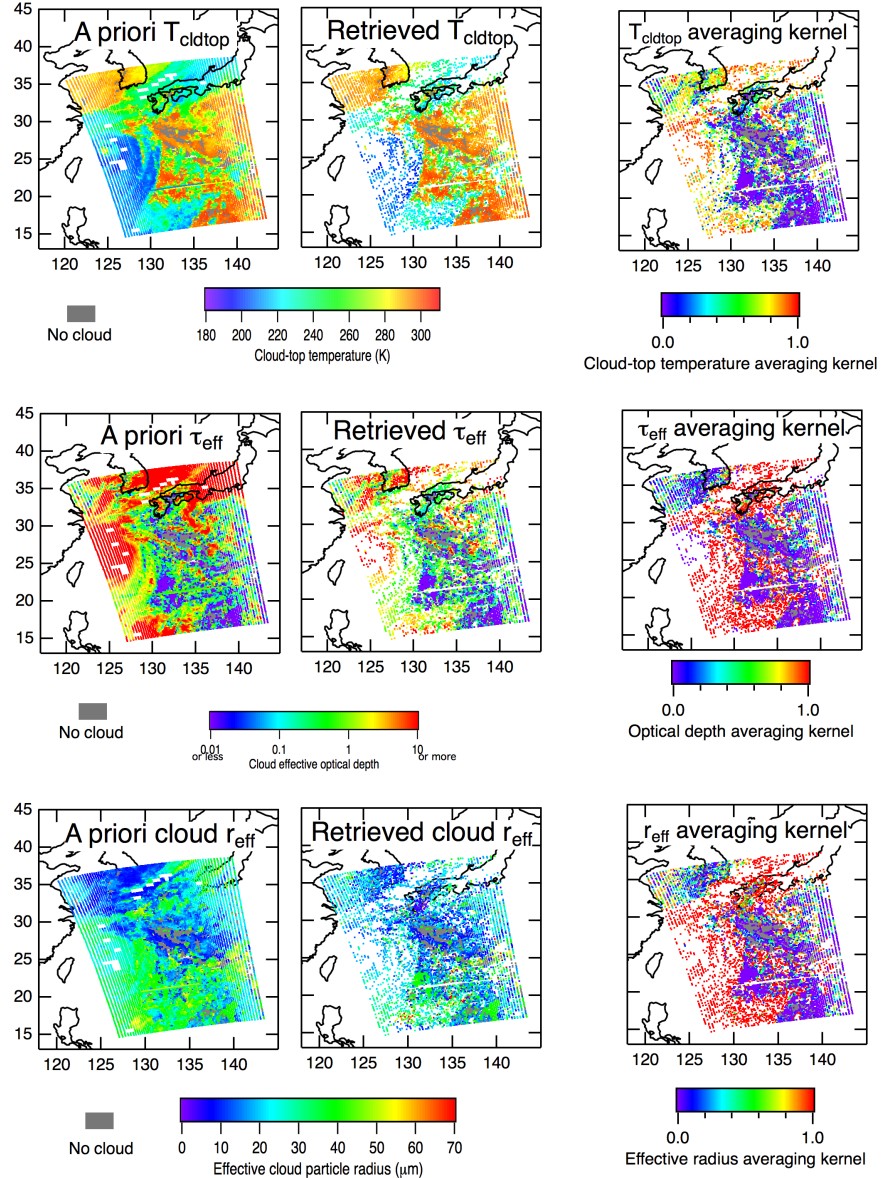

**Figure 6:** Sample a priori, AIRS-OE retrievals and AIRS-OE averaging kernels of cloud-top temperature ($T_{cldtop}$), effective optical depth ($\tau_{eff}$) and effective particle radius ($r_{eff}$). Data are from AIRS (daytime) Granule 44, September 6, 2002.



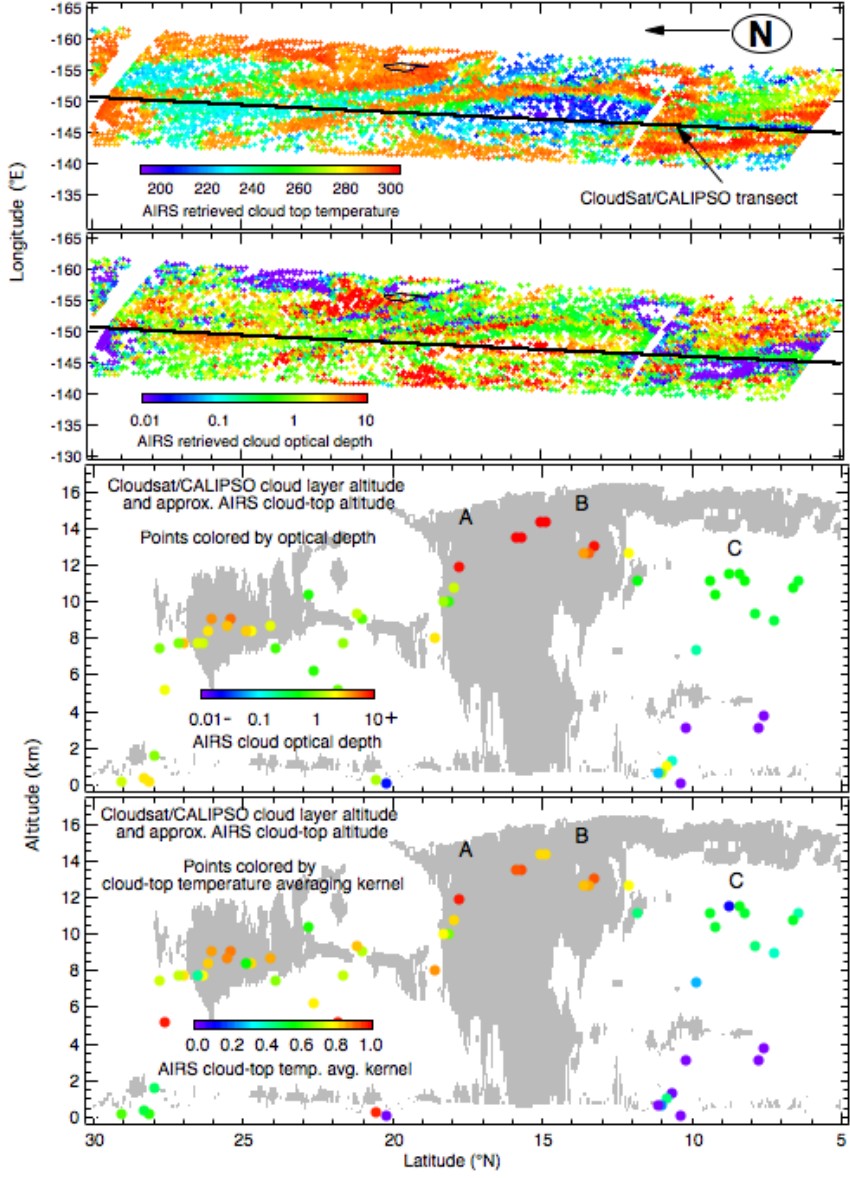

**Figure 7:** Comparison of AIRS-OE approximate cloud-top altitude with CloudSat/CALIPSO (CsC). Top panel: AIRS-OE cloud-top temperature with CsC transect. Second panel: AIRS-OE cloud optical depth. Third panel: CsC cloud layer altitudes (grey) with approximate AIRS-OE cloud-top altitudes, colored by retrieved optical depth. Bottom panel: Same as third panel, but points colored by cloud-top temperature averaging kernels. Data are from daytime observations, July 31, 2009 over the Pacific ocean.




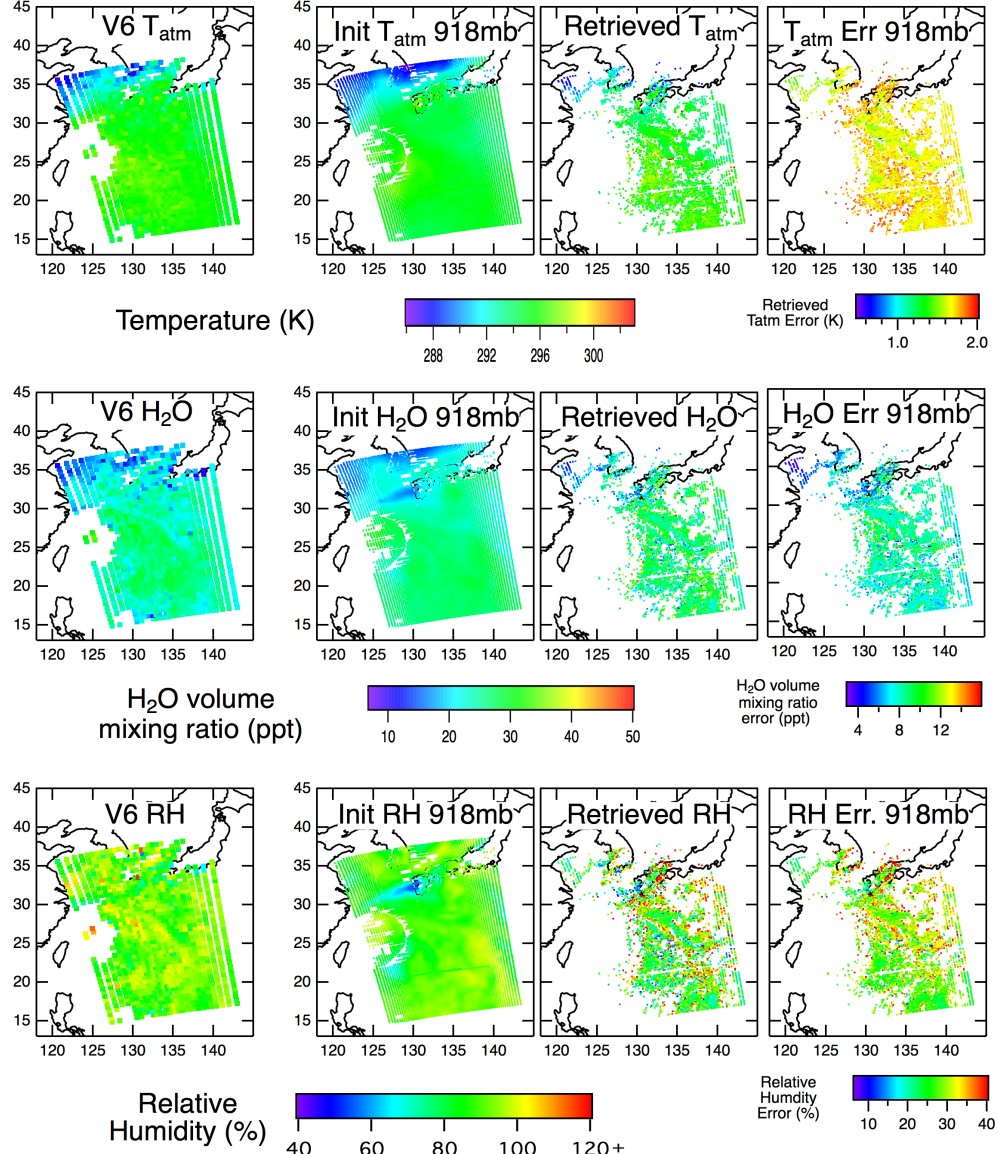

**Figure 8:** AIRS V6 retrieval, and AIRS-OE a priori, retrievals and errors for temperature, water vapor and relative humidity (RH) at 918 mb for (daytime) Granule 44, September 6, 2002.





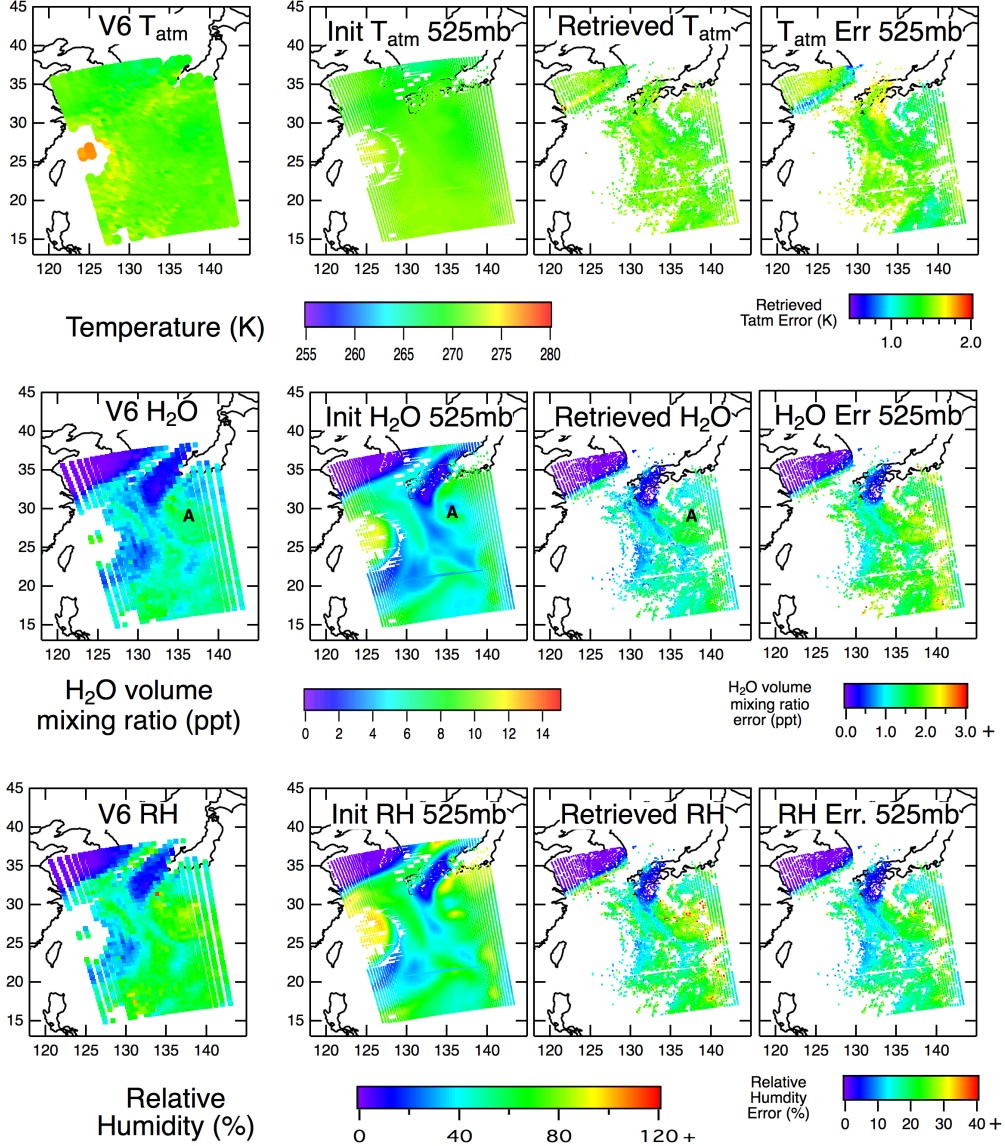

**Figure 9:** AIRS V6 retrieval, and AIRS-OE a priori, retrievals and estimated errors for temperature, water vapor and relative humidity (RH) at 525 mb for (daytime) Granule 44, September 6, 2002.



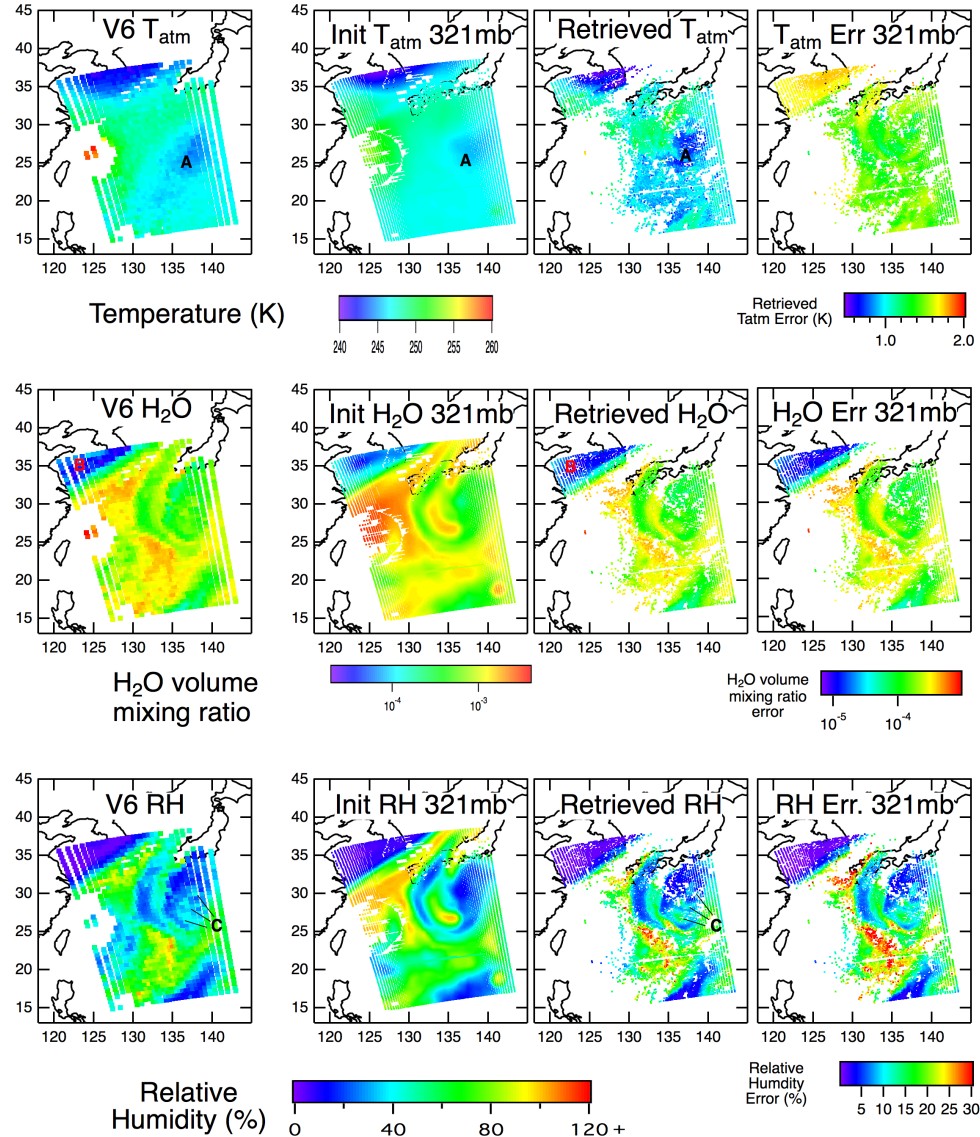

**Figure 10:** AIRS V6 retrieval, and AIRS-OE a priori, retrievals and errors for temperature, water vapor and relative humidity (RH) at 321 mb for (daytime) Granule 44, September 6, 2002. Note that unlike Figures 8 and 9, water vapor mixing ratio is on a logarithmic scale.



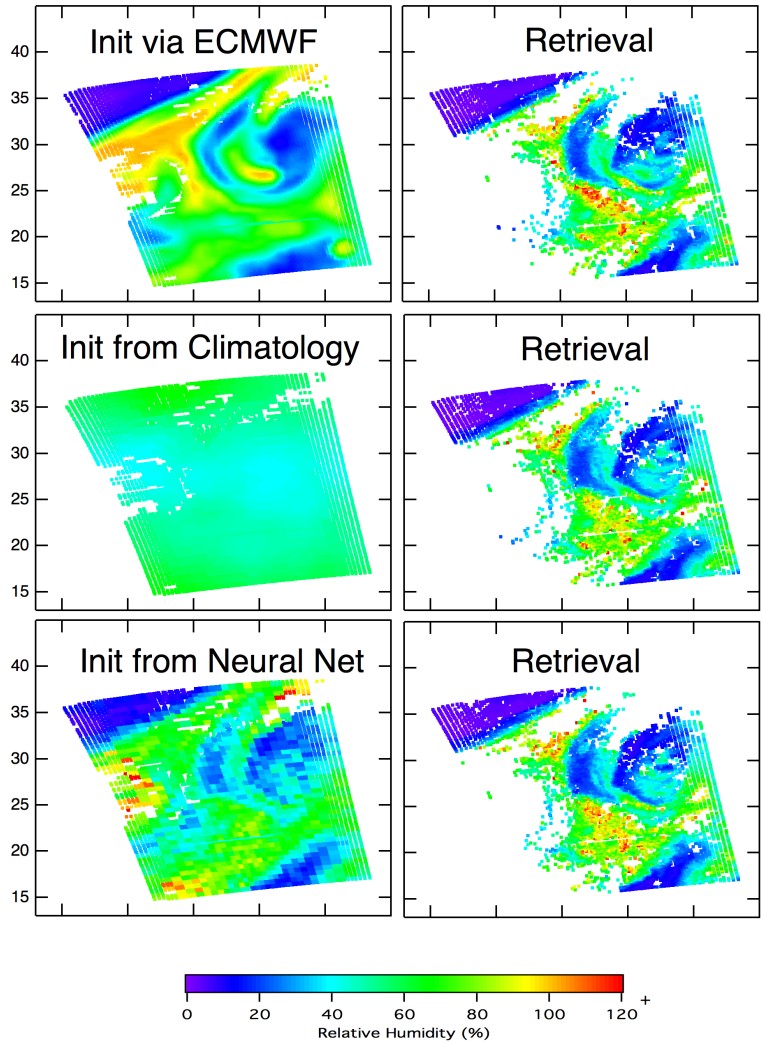

**Figure 11:** AIRS-OE relative humidity (calculated from retrieved temperature and water vapor) at 318 mb using different a priori. The left panels show the relative humidity calculated from the a priori, while the right panels show the retrieval. The top row uses an ECMWF analysis-derived a priori linearly interpolated in time, space and log pressure to the AIRS observation. The second row uses a climatology. The third row uses the neural-net calculation (on the AIRS-AMSU footprint) that is used with the operational AIRS V6 retrieval. Note that while the different a priori were used for temperature, skin temperature and water vapor, the same MODIS-derived cloud data (cloud-top temperature, cloud optical depth and cloud particle radius) were used as a priori for the cloud retrievals.





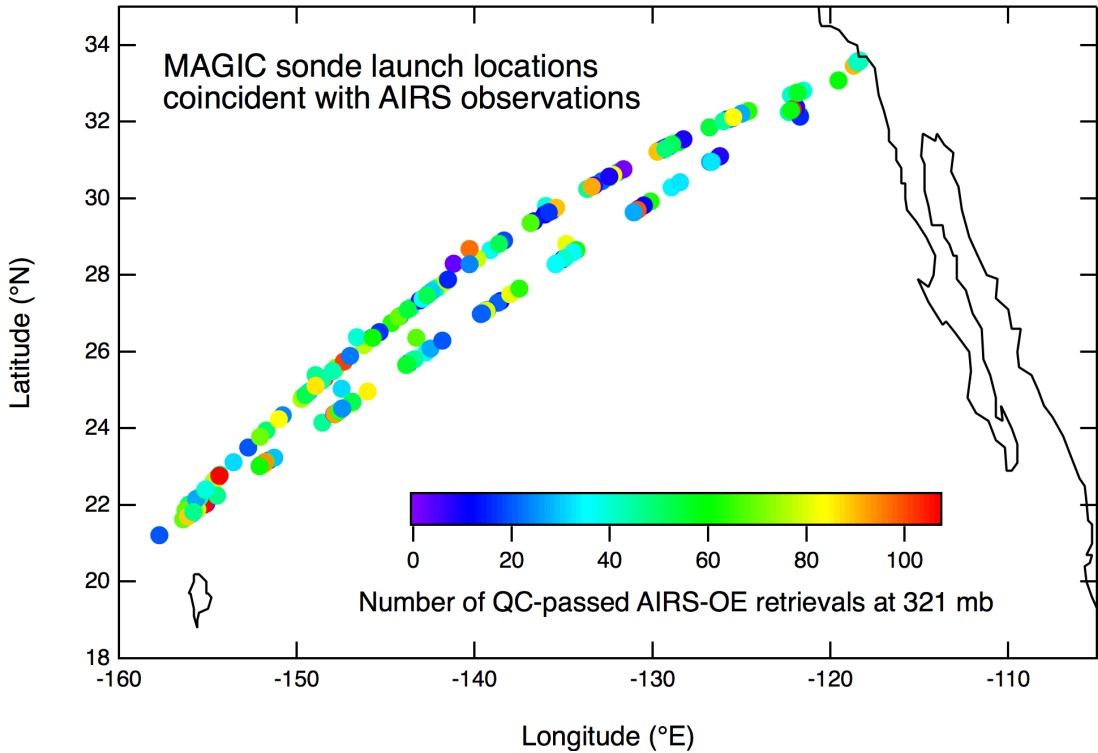

**Figure 12:** MAGIC sonde launch locations that were matched to coincident AIRS observations. Points are colored by the number of QC-passed AIRS-OE retrievals of water vapor at 321 mb. AIRS observations were within 3 hours and 100 km of sonde launch.





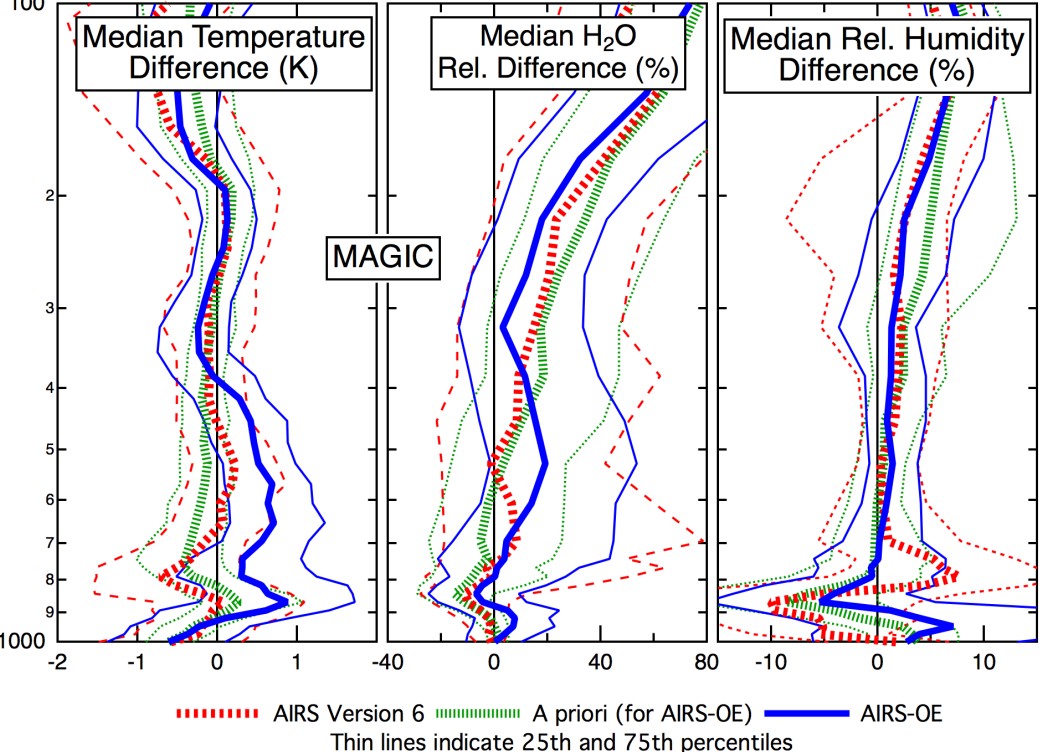

**Figure 13:** AIRS-OE, AIRS-Version 6 and a priori profile biases from MAGIC radiosondes. Left panel: Median temperature difference (minus sonde). Middle panel: median relative difference in water vapor ([AIRS – sonde] / sonde in %). Right panel: Median difference in relative humidity (in %). Thin lines represent 25[th] and 75[th] percentiles. AIRS observations were within 3 hours and 100 km of sonde launch, and only those retrieval layers passing quality control were counted. In total, 7879 AIRS profiles were matched to 210 sonde profiles. See Sec. 5.1 for description of data aggregation before calculation of medians.





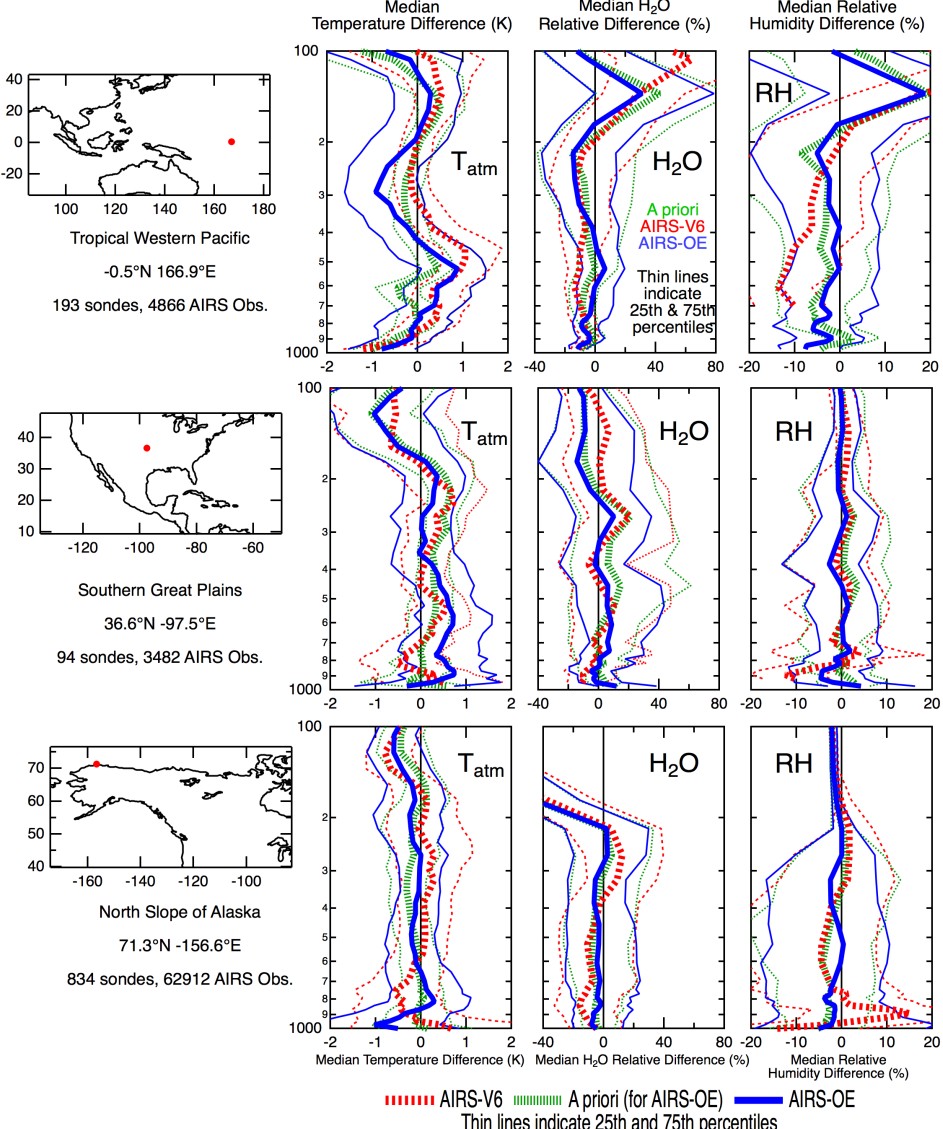

**Figure 14:** AIRS-OE, AIRS-Version 6 and (AIRS-OE) a priori profile differences from radiosondes launched from the Tropical Western Pacific site (top row), Southern Great Plains (middle row), and North Slope of Alaska (bottom row). Left column: Median temperature differences (minus sonde). Middle column: median relative difference in water vapor ([AIRS − sonde] / sonde in %). Right column: median difference in relative humidity (in %). Thin lines represent 25th and 75th percentiles. AIRS observations were within 3 hours and 100 km of sonde launch and only those retrieved layers passing quality control were counted. The a priori for AIRS-V6 is not shown. See Sec. 5.1 for description of data aggregation before calculation of medians.