# Peer review of "Single-footprint retrievals of temperature, water vapor and cloud properties from AIRS"

_Atmospheric Measurement Techniques, 2017_

## Referee Comment (RC1) · Anonymous Referee #1 · 22 Aug 2017

General comments:

The authors describe a 1-D variational retrieval of temperature, atmospheric constituent gases, and cloud properties from AIRS observations (AIRS-OE) and co-located operational MODIS cloud property retrievals. Multiple retrieval strategies are employed based on the availability of MODIS cloud properties and their sub-pixel characteristics within an AIRS FOV. The paper fits well within the scope of AMT and provides a sufficient contribution to scientific progress in the field of remote sounding from hyperspectral IR measurements.

The layout of the paper is logical and the algorithm flow is well described; however, there are significant technical and material deficiencies that need addressed before acceptance to AMT.

[Figure]

For instance, some important details of algorithm settings (e.g., measurement noise) are missing from the text and the authors' description of the retrieval information content/error estimation is sometimes confusing. In addition, the sections on the discussion of results, validation of the AIRS-OE algorithm against matched radiosonde, and comparisons to operational AIRS v6 (and conclusions drawn from the results) are lacking enough detail to fully assess the comparative pros, cons, and skill of the AIRS-OE retrieval relative to existing products.

Specific comments:

"QA" is not defined (pages 13, 14, 15) in the text.

Page 1, Line 20: "higher vertical resolution of retrieved temperature and water vapor" There are other advantages of thermal infrared data – e.g., trace gas sensitivities, sensitivity to aerosol, . . .

Page 2, Line 10: . . . the Stand-alone AIRS Radiative Transfer Algorithm, Delta-Four Stream (SARTA-D4S; Ou et al., 2013). There is another extension to the SARTA package that enables the simulation of outgoing radiance in the presence of cloud and aerosol and that is directly applicable to AIRS studies: DeSouza-Machado, S., Strow, L. L., Imbiriba, B., McCann, K., Hoff, R., Hannon, S., Martins, J., Tanré, D., Deuzé, J., Ducos, F., and Torres, O.: Infrared retrievals of dust using AIRS: comparisons of optical depths and heights derived for a North African dust storm to other collocated EOS A-Train and surface observations, J. Geophys. Res., 115, D15201, doi:10.1029/2009JD012842, 2010.

Page 4, Line 2: How is the radiance noise covariance prescribed? AIRS detector modules have significant correlated noise among channels within each module (e.g., http://onlinelibrary.wiley.com/doi/10.1256/qj.03.93/pdf., page 1480 and Tobin, et al., "Hyperspectral data noise characterization using principal component analysis," J. Appl. Remote Sensing, v1, 2006). Does the algorithm account for the correlation in the observation covariance, or does it only use a diagonal matrix? How is the noise

estimated or prescribed? Are forward model error components estimated or used in addition to instrument noise? If so, how? Do you use a bias correction between observations minus calculations; if so, how is it estimated?

Page 5, line 35: "use extrapolated cloud absorption and scattering parameters, and may not be reliable." This statement is not clear. I assume that the cloud absorption and scattering parameters are restricted in the Forward Model lookup tables between 5-85 microns, but it is not clear from the text.

Page 8, line 2: "...of uncertainties in the scattering/absorption ratio (Nakajima and King, 1990; Nakajima et al., 1991)" ... perhaps also due to the 4 stream approximation of the forward model employed in this study.

Page 8, Line 13: "...First, leaving other variables fixed, only $\tau$cld is retrieved...The resultant spectral fit from this initial retrieval can be poor." If the spectral fits are poor after this first attempt how is convergence/iteration stopping criteria determined? This particular detail of the algorithm flow seems important and critical to the success or failure of the algorithm to converge to the optimal solution in subsequent steps.

Page 8, Line 23: "...nearly linear in vicinity of the solution" and a priori? Per the above comment about poor spectral fits after the first attempt, I wonder how well this linear assumption holds for all retrieval strategies employed. The authors should address the potential failure of this assumption and/or reasons why the assumption is valid in greater depth.

Page 9, Line 33: "...on the spectrum" suggest to revise to "measurement spectrum."

Pages 9-10: How does the vertical resolution of the AIRS-OE algorithm compare to the operational AIRS V6 vertical resolution.

Page 10, Line 12-13: Continuing the previous comment. "... depend on the amounts of trace gases present, the temperature lapse rate, the particulars of the cloud field" These statements are true for any retrieval strategy (cloud-clearing, cloudy, or clear)

from AIRS or any other instrument. The authors continue "...since the AIRS-OE retrievals are simultaneous and not sequential...," which seems to be a nod to the operational AIRS algorithm. Consider adding a reference (e.g., Maddy, E.S. and C.D. Barnet 2008. Vertical resolution estimates in Version 5 of AIRS operational retrievals. IEEE TGRS. v.46 Section 2A, p. 2377) to back up the first statement and provide contrast the second statement.

Page 10, Line 25: How is the noise error covariance defined? See above.

Page 10, Lines 29-30): "...spectral biases or other errors that are correlated across observations" calibration uncertainty, correlated instrument noise (if not included in measurement covariance . . . see previous comments).

Page 12, Line 23-25: "...The morphology of the AIRS-OE retrieval fields are similar to the a priori, and the morphology of the averaging kernel fields are similar to each other. " This statement is confusing. Please revise for clarity..

Page 13, Line 26: "...for each of these..." Consider revising to "for each retrieval quantity"

Page 13, Line 30: "...unphysically high values..." It's unclear whether the authors are referring to unphysically high differences (>100%) or unphysically super-saturated retrievals.

Section 4.3: Spatial resolution is one of the main differences between AIRS-OE and AIRS V6. Have the authors performed a comparison between quality controlled AIRS-OE averaged onto the AIRS V6 effective footprint (i.e., 3x3 spatial average)? A short discussion of this type of comparison is suggested.

Page 13, Line 33: "IGR" should be IQR.

Page 14, Lines 11-16: Authors provide a qualitative description of the "a priori" sensitivity of the algorithm. A more detailed quantitative would be informative to the reader.

[Figure]

Page 14, Line 30: "...lower down..." consider removing.

Page 14, Line 33: "...because the water vapor retrieval was too high..." Too high relative to what? Consider revising for clarity. For instance, "because the reported retrieval water vapor was supersaturated."

Figures 13, 14 and corresponding text: The AIRS-OE a priori statistics (bias and IQR) are generally more well behaved as compared to both the AIRS v6 and AIRS-OE retrievals and in some cases perform better than both physical retrieval algorithms (esp. near surface for temperature). I wonder how much do the cloud/other variable retrievals compensate for other variable/cloud errors. A more detailed analysis and/or discussion of an assessment of AIRS-OE retrieval increments of profile variables as well as surface temperature, and cloud optical properties relative to a priori as compared to Radiosonde and correlative CloudSat/Calipso retrievals should elucidate cross-talk between retrieval parameters is suggested.

Page 15, line 30: "are low" – how low?

Page 16, line 9: "...has an information content analysis ... operates both within and across different atmospheric parameters." It is unclear what is meant by "operates both within and across." Do you mean that the information content analysis provides diagnostic information regarding the temperature retrieval, water retrieval, etc. and interactions between temperature and water, temperature and cloud, water and cloud, etc. If so, please revise.

Page 16, line 16-17: " ... incorporating scattering by dust ..." Is the radiative effect of an atmospheric dust signal in AIRS measurements large enough to cause significant degradation in a 1DVAR retrieval? Consider adding a sentence and or reference detailing why this is might be important (e.g., Maddy, E., DeSouza-Machado, S., Nalli, N., Barnet, C., Strow, L., Wolf, W., Xie, H., Gambacorta, A., King, T., Joseph, E., Morris, V., Hannon, S., and Chou, P.: On the effect of dust aerosols on AIRS and IASI operational level 2 products, GRL, 39, L10809, doi:10.1029/2012GL052070, 2012. 446)

[Figure]

Page 16, line 17-18: Again, does the 4-stream RT algorithm limit the use of AIRS shortwave channels? Are there any other specific limitations with forward modeling that need to be addressed before the 4micron band could be utilized? Scattering tables, spectroscopy, etc.?

Page 16, line 32: "...compare well with operational AIRS-V6..." how well? how is it distinguished from AIRS v6? There are a number of characteristics (potential advantages and distinguishing features) of the AIRS-OE algorithm as compared to the AIRS v6 algorithm (higher spatial and vertical resolution, potentially better cloud characterization, comparable statistical results, etc.); however, these are scattered within the text and conclusions. It would be helpful to re-organize the conclusions such that these characteristics are emphasized.

---

## Referee Comment (RC2) · Anonymous Referee #2 · 25 Sep 2017

Overall assessment. This manuscript tackles one of the most fundamental issues in the field of hyperspectral IR remote sensing, that is the problem of clouds in an inversion scheme. The text is very well written and provides a way forward to the advancement of the inversion problem. All assumptions in the treatment of the cloud a priori are well explained. This lays out the foundation for a constructive conversation on the future directions that IR retrieval developers may want to consider when it comes to address the presence of clouds in an inversion methodology. I suggest this manuscript to be published, pending minor corrections as outlined below.

Minor comments: 1. The authors should add that, besides not needing absorption and scattering in the forward model, another advantage of cloud clearing is that it also does not require a formal cloud geophysical a priori and its error co-variance; it is

[Figure]

computationally fast; it enables full column retrievals. A limitation, besides assuming constant water vapor in the cluster of FOVs, comes from also assuming uniform surface properties over the cluster of FOVs, which can be challenging over coastal regions.

2. Could the authors explain in few sentences how their work differs from the cited existing methods employed for direct use of cloudy infrared spectra in atmospheric retrievals (Liu, 2009, Blackwell 2005 and Kulawik, 2006a).

3. Can the authors say anything about using a monthly averaged surface emissivity, is this a robust approximation for a given day of the month? Also, have they evaluated the inter-annual variability in the data set to derive a rough assessment of how good of an approximation is to apply the 2003 climatology to the year before? Have they attempted a comparison with the existing surface emissivity regression solution of AIRS v6? If not, can they say few words about their motivation to replace the AIRS v6 method? Also, for completeness, they should include this as part of the list of differences with respect to AIRS v6 on page 2 -3.

4. Do the MODIS cloud parameters come with a formally computed error covariance? Otherwise, what is the source for this?

5. Was there a specific geophysical regime where the use of the 9.6 micron band was problematic, for example the tropical region, or desert areas? The broader scientific community would benefit from a more detailed explanation of what the sentence "We therefore retrieve O3 only as an "interferent" gas within the 14 $\mu$m CO2 region, and avoid the 9.6 $\mu$m band" means. Did the authors only use channels in the CO2 band for the retrieval of ozone? If not, can they explain more?

6. Section 3.6 Is the approximation of leaving all other variables fixed taken into account in the retrieval equation, for example, similarly to what is done in AIRS v6? Also, in the list of differences from the AIRS v6 approach, the authors stated that this is a simultaneous retrieval approach. Can they more fairly say that this is a two-step sequential approach?

7. Also about section 3.6, if in within the page limits, can the authors say few words on the convergence criteria, besides mentioning the reference to Bowmann et al, 2006? Are those the same as outlined later in the text? If yes, for clarity, the authors should refer the reader to that section.

8. What enters in the measurement noise covariance term, only instrument noise? Can the authors specify this?

9. As an alternative method to the median of the medians, the authors should try the methodology described in Nalli et al., "Validation of satellite sounder environmental data records: Application to the Cross-track Infrared Microwave Sounder Suite", JGR, 2013, where each layer statistic is weighted by the mean water vapor quantity in the layer. This method was also used in Tobin et al., 2005, JGR and seems to be a well established methodology in the satellite retrieval field.

10. Summary and discussion section. How are the authors planning to extend this retrieval method to multi-layer cloud retrievals. Does the MODIS retrieval output contain multi-layer clouds that can be used as a priori?

11. Could the authors also consider comparing their cloud retrieval products to the AIRS v6 cloud retrieved products?

Technical comments: 1. The broader science community might not be familiar with what "L1b" data are. The author should either keep using the "Level 1" definition for radiance data or explain what the "b" stands for.

---

## Author Comment (AC1) · 30 Nov 2017

In the revision of our paper, we have added more detail per the reviewer's concerns. Most importantly, we made a significant change to the retrieval algorithm in that we no longer have the initial, "tentative" retrieval of cloud optical depth ($\tau_{cld}$) prior to the simultaneous retrieval of $\tau_{cld}$, $T_{cldtop}$, $r_{eff}$, $T_{sfc}$, $T_{atm}$, $H_2O$, $O_3$ and $CO_2$. Figures and quantitative comparisons to radiosondes, AIRS-V6 and CloudSat/CALIPSO are updated. Details of this and other specific areas are addressed in the responses below.

*General comments:*

*The authors describe a 1-D variational retrieval of temperature, atmospheric constituent gases, and cloud properties from AIRS observations (AIRS-OE) and co-located operational MODIS cloud property retrievals. Multiple retrieval strategies are employed based on the availability of MODIS cloud properties and their sub-pixel characteristics within an AIRS FOV. The paper fits well within the scope of AMT and provides a sufficient contribution to scientific progress in the field of remote sounding from hyper- spectral IR measurements.*

*The layout of the paper is logical and the algorithm flow is well described; however, there are significant technical and material deficiencies that need addressed before acceptance to AMT.*

*For instance, some important details of algorithm settings (e.g., measurement noise) are missing from the text and the authors' description of the retrieval information content/error estimation is sometimes confusing. In addition, the sections on the discussion of results, validation of the AIRS-OE algorithm against matched radiosonde, and comparisons to operational AIRS v6 (and conclusions drawn from the results) are lacking enough detail to fully assess the comparative pros, cons, and skill of the AIRS-OE retrieval relative to existing products.*

*Specific comments:*

*"QA" is not defined (pages 13, 14, 15) in the text.*

We've changed "QA" to "QC", and spell out this abbreviation in Sec 3.8.

*Page 1, Line 20: "higher vertical resolution of retrieved temperature and water vapor" There are other advantages of thermal infrared data – e.g., trace gas sensitivities, sensitivity to aerosol, . . .*

We agree, but trace gases were retrieved here only insofar as they improved the temperature, water vapor and cloud retrievals. We didn't get into the advantages of the infrared with respect to trace gases or aerosol because we (mostly) did not retrieve them in this initial effort. We have changed the phrase "The advantage …" to "An advantage …"

*Page 2, Line 10: . . . the Stand-alone AIRS Radiative Transfer Algorithm, Delta-Four Stream (SARTA-D4S; Ou et al., 2013). There is another extension to the SARTA package*

*that enables the simulation of outgoing radiance in the presence of cloud and aerosol and that is directly applicable to AIRS studies: DeSouza-Machado, S., Strow, L. L., Imbiriba, B., McCann, K., Hoff, R., Hannon, S., Martins, J., Tanré, D., Deuzé, J., Ducos, F., and Torres, O.: Infrared retrievals of dust using AIRS: comparisons of optical depths and heights derived for a North African dust storm to other collocated EOS A-Train and surface observations, J. Geophys. Res., 115, D15201, doi:10.1029/2009JD012842, 2010.*

We didn't reference the DeSouza-Machado et al. (2010) paper since they it was mostly about dust retrieval (something we didn't try in AIRS-OE). However, the referee is correct in that it was a significant extension to the forward modelling in AIRS, and this is noted in the conclusions. We have also made note of the recent DeSouza-Machado et al. paper currently in discussion in AMT.

*Page 4, Line 2: How is the radiance noise covariance prescribed? AIRS detector modules have significant correlated noise among channels within each module (e.g., http://onlinelibrary.wiley.com/doi/10.1256/qj.03.93/pdf., page 1480 and Tobin, et al., "Hyperspectral data noise characterization using principal component analysis," J. Appl. Remote Sensing, v1, 2006). Does the algorithm account for the correlation in the observation covariance, or does it only use a diagonal matrix? How is the noise estimated or prescribed? Are forward model error components estimated or used in addition to instrument noise? If so, how? Do you use a bias correction between obser- vations minus calculations; if so, how is it estimated?*

Only the diagonal error is reported in the AIRS Level 1b data, and we have not estimated off-diagonal elements. This should have been noted, and we thank the reviewer for pointing this out. We have added some comments about this, and a reference to the Tobin et al. paper. We have not put in estimates of the systematic error components for the forward model (this was noted in 3.7.2), but we expanded on this a little and w.r.t. bias corrections, and we note tuning done in SARTA and refer to the validation paper by Strow et al. (2006). We now note that random error from the "clear" part of the forward model is expected to be less than random errors from the observed radiance, but it should be included in future versions.

*Page 5, line 35: "use extrapolated cloud absorption and scattering parameters, and may not be reliable." This statement is not clear. I assume that the cloud absorption and scattering parameters are restricted in the Forward Model lookup tables between 5-85 microns, but it is not clear from the text.*

We have changed to line to read, "Lookup tables for cloud absorption and scattering parameters had a particle radii range from 5 to 85 μm; reported cloud absorbtion and scattering parameters outside this range rely on extrapolated parameters, and results may not be reliable."

*Page 8, line 2: "...of uncertainties in the scattering/absorption ratio (Nakajima and King, 1990; Nakajima et al., 1991)" . . . perhaps also due to the 4 stream approximation of the forward model employed in this study.*

This is possible, but we also encountered difficulties in clear scenes when using the shortwave channels, and so we think it likely that the problems of reflected sunlight contribute substantially to this. Whether this is an issue with just SARTA, or with D4S as well needs investigation. We also did not adjust the noise in the shortwave channels as a function of channel radiance, as illustrated in the Tobin et al. paper. As we used the L1b errors directly (which tend to be less than those of cloud-cleared radiances in the shortwave), we tended to rely more strongly on the accuracy of the forward model in the shortwave (and the fidelity of the reflected sunlight calculation) than would be done in AIRS-V6. Profile biases and failed retrievals when using the shortwave were non-trivial problems that we could not quickly solve, so we decided at this stage of development to avoid the issue altogether and just use longwave channels.

*Page 8, Line 13: "…First, leaving other variables fixed, only τcld is retrieved…The resultant spectral fit from this initial retrieval can be poor." If the spectral fits are poor after this first attempt how is convergence/iteration stopping criteria determined? This particular detail of the algorithm flow seems important and critical to the success or failure of the algorithm to converge to the optimal solution in subsequent steps.*

We note that a retrieval can "converge" such that subsequent interations would not lower the cost function any further; the fit can be poor (as measured by the chiSquare), but better than one calculated directly from the a priori, and that we did not change $x_0$ in the cost function or the a priori covariance of $\tau_{cld}$ from the first step to the simultaneous step – perhaps avoiding the warning in Sec. 5.6.2 of Rodgers [2000]. Nonetheless, we reviewed this aspect of the algorithm, and reprocessed validation data from the TWP, SGP and NSA sites, skipping the intitial retrieval of cloud optical depth and going directly to the simultaneous retrieval. We found that to get close to the earlier yields, we had to double the number of allowable iterations.  The graph below is a histogram of the number of iterations needed with and without the initial retrieval of cloud optical depth over TWP.

[Figure]

Comparisons against radiosondes (using the "median-of-the-medians" approach) shows effectively no differences at NSA, some differences at SGP, and the largest differences at TWP. The TWP comparisons (below) show small and isolated biases with each other, but we think they are significant enough to call into question whether the initial retrieval of cloud optical depth can lead to an optimal solution.

[Figure]

"Median of medians" Differences between AIRS-OE retrievals and sondes at TWP

With initial retrieval of $\tau_{cld}$ (30 iterations max)
Without initial retrieval of $\tau_{cld}$ (60 iterations max)
Dotted lines show 25th and 75th percentiles

Given this, and despite the extra processing required, we decided to remove the initial retrieval of cloud optical depth, and reprocess the AIRS data shown in the paper. In reflection of this, we have re-written Sec. 3.6, and modified the block diagram of Figure 2. The retrieval that produced the averaging kernels and row sums in Figures 4 and 5 of the original draft failed under this new processing, so we selected a new retrieval to produce the sample graphics. We thank the reviewer for bringing up this issue and prompting us to study it further.

*Page 8, Line 23: "…nearly linear in vicinity of the solution" and a priori? Per the above comment about poor spectral fits after the first attempt, I wonder how well this linear assumption holds for all retrieval strategies employed. The authors should address the potential failure of this assumption and/or reasons why the assumption is valid in greater depth.*

The assumption of linearity is perhaps most important in the error characterization. As the a priori covariances are largely ad hoc, we cautioned against the errors being reliable. We thought it best to leave validation of the error characterization, and the linearity assumptions that go with that, to a later study when AIRS-OE is more

refined. We have added a phrase noting that the assumption of linearity has not been tested. In the Summary and Discussion section, we mention adding linearity tests to future versions and point the reader to the section in Rodgers [2000] that expands on this.

*Page 9, Line 33: "...on the spectrum" suggest to revise to "measurement spectrum."*

Fixed. It now reads "...on the measured spectrum..."

*Pages 9-10: How does the vertical resolution of the AIRS-OE algorithm compare to the operational AIRS V6 vertical resolution.*

We've have not tested this specifically as it goes beyond the scope of this initial effort. Making an fair comparison between vertical resolution estimates between the two versions has to be performed and documented carefully as the averaging kernels of AIRS-OE and AIRS-V6 (the latter using singular value decomposition) are calculated quite differently.

*Page 10, Line 12-13: Continuing the previous comment. "... depend on the amounts of trace gases present, the temperature lapse rate, the particulars of the cloud field" These statements are true for any retrieval strategy (cloud-clearing, cloudy, or clear) from AIRS or any other instrument.*

True, but the section walks the reader through a sample averaging kernel for a simultaneous retrieval, and explains how its information can be extracted and used. We thought it helpful to tell the reader less familiar with averaging kernels that they're quite scene-specific.

*The authors continue "...since the AIRS-OE retrievals are simultaneous and not sequential...," which seems to be a nod to the operational AIRS algorithm. Consider adding a reference (e.g., Maddy, E.S. and C.D. Barnet 2008. Vertical resolution estimates in Version 5 of AIRS operational retrievals. IEEE TGRS. v.46 Section 2A, p. 2377) to back up the first statement and provide contrast the second statement.*

Section 3.7.1 now references the Maddy and Barnet paper, but we note that add that the averaging kernels from the AIRS operational retrievals do not work *across* species.

*Page 10, Line 25: How is the noise error covariance defined? See above.*

Section 3.3 has been expanded to describe this more fully, and we refer back to that section.

*Page 10, Lines 29-30): "...spectral biases or other errors that are correlated across observations" calibration uncertainty, correlated instrument noise (if not included in measurement covariance . . . see previous comments).*

The lines relate that we are not calculating systematic error in Eq. 11, and systematic error would be affected by calibration uncertainty. The correlated noise

would more affect random error, and we have noted that it's not there (but eventually will be) in the line below Eq. 10. We added a note that Strow et al. use tuning to fix some of the systematic error coming out of SARTA.

*Page 12, Line 23-25: ". . .The morphology of the AIRS-OE retrieval fields are similar to the a priori, and the morphology of the averaging kernel fields are similar to each other. " This statement is confusing. Please revise for clarity..*

We're re-written this to read: "Retrieved quantities have similar fields to their a priori. As indicated by the averaging kernels, sensitivity is enhanced for ice clouds, which is likely because of the higher thermal contrast with the surface."

*Page 13, Line 26: "...for each of these..." Consider revising to "for each retrieval quantity"*

Done. Thanks.

*Page 13, Line 30: "...unphysically high values..." It's unclear whether the authors are referring to unphysically high differences (>100%) or unphysically super-saturated retrievals.*

We've changed to read "...unphysically high RH values..."

*Section 4.3: Spatial resolution is one of the main differences between AIRS-OE and AIRS V6. Have the authors performed a comparison between quality controlled AIRS-OE averaged onto the AIRS V6 effective footprint (i.e., 3x3 spatial average)? A short discussion of this type of comparison is suggested.*

We agree that such a comparison would be interesting, but this is beyond the scope of the current study. Such an analysis would be useful after we amelorate many of the retrieval's listed shortcomings and move to a more mature code.

*Page 13, Line 33: "IGR" should be IQR.*

Fixed. Thanks.

*Page 14, Lines 11-16: Authors provide a qualitative description of the "a priori" sensitivity of the algorithm. A more detailed quantitative would be informative to the reader.*

We're adding in a profile plot of the average row sums of the Tatm and H2O averaging kernels for retrievals using the ECMWF a priori. Plots using the climatology or neural-net a priori were very similar.

*Page 14, Line 30: ". . .lower down. . ." consider removing.*

Removed.

*Page 14, Line 33: "...because the water vapor retrieval was too high..." Too high relative to what? Consider revising for clarity. For instance, "because the reported retrieval water vapor was supersaturated."*

We've changed the text to read, "We also found that average bias could often be significantly skewed by retrieval outliers – usually because the water vapor retrieval and relative humidity were unphysically high."

*Figures 13, 14 and corresponding text: The AIRS-OE a priori statistics (bias and IQR) are generally more well behaved as compared to both the AIRS v6 and AIRS-OE retrievals and in some cases perform better than both physical retrieval algorithms (esp. near surface for temperature). I wonder how much do the cloud/other variable retrievals compensate for other variable/cloud errors. A more detailed analysis and/or discus- sion of an assessment of AIRS-OE retrieval increments of profile variables as well as surface temperature, and cloud optical properties relative to a priori as compared to Radiosonde and correlative CloudSat/Calipso retrievals should elucidate cross-talk between retrieval parameters is suggested.*

We agree that more detailed analyses would be useful, but with the resources given this effort, we could only make limited comparisons against sondes and comparisons to AIRS-V6. As noted in the Summary and Discussion section, there are several non-trivial liens that we have identified against the algorithm that should be fixed for the upcoming versions. A more comprehensive validation and analysis would be warranted as the algorithm matures.

*Page 15, line 30: "are low" – how low?*

We've changed the line to read: "At NSA between about ~800 and ~220 mb, AIRS-OE temperature biases are low (less than ±0.25 K) at altitudes above ~800 mb, as are water vapor relative biases (less than ±6%)"

*Page 16, line 9: ". . .has an information content analysis . . . operates both within and across different atmospheric parameters." It is unclear what is meant by "operates both within and across." Do you mean that the information content analysis provides diagnostic information regarding the temperature retrieval, water retrieval, etc. and interactions between temperature and water, temperature and cloud, water and cloud, etc. If so, please revise.*

We changed the sentence to add examples: "As AIRS-OE rests on an optimal estimation framework, and includes simultaneous retrieval of profiles and scalar variables, it has an information content analysis that operates both within an atmospheric parameter (e.g., uncertainties in the temperature profile) and across different atmospheric parameters (e.g., uncertainties in water vapor due to uncertainties in temperature)."

*Page 16, line 16-17: ". . . incorporating scattering by dust . . ." Is the radiative effect of an atmospheric dust signal in AIRS measurements large enough to cause significant degradation in a 1DVAR retrieval? Consider adding a sentence and or reference de- tailing why this is might be important (e.g., Maddy, E., DeSouza-Machado, S., Nalli, N., Barnet, C., Strow, L., Wolf, W., Xie, H., Gambacorta, A., King, T., Joseph, E., Morris, V., Hannon, S., and Chou, P.: On the effect of dust aerosols on AIRS and IASI operational level 2 products, GRL, 39, L10809, doi:10.1029/2012GL052070, 2012. 446)*

Done. Thanks.

*Page 16, line 17-18: Again, does the 4-stream RT algorithm limit the use of AIRS shortwave channels? Are there any other specific limitations with forward modeling that need to be addressed before the 4micron band could be utilized? Scattering tables, spectroscopy, etc.?*

As noted above, these are non-trivial issues with SARTA (or most any other thermal-IR forward model). At this stage, we hesitate to speculate on exactly what changes are needed in SARTA, or SARTA-D4S to fix issues in the shortwave, although we note modifying the radiance error may help in the retrieval.

*Page 16, line 32: "…compare well with operational AIRS-V6…" how well? how is it distinguished from AIRS v6? There are a number of characteristics (potential advantages and distinguishing features) of the AIRS-OE algorithm as compared to the AIRS v6 algorithm (higher spatial and vertical resolution, potentially better cloud characterization, comparable statistical results, etc.); however, these are scattered within the text and conclusions. It would be helpful to re-organize the conclusions such that these characteristics are emphasized.*

We list what we think are the most useful and distinguishing features of AIRS-OE compared to AIRS-V6 in the first paragraph of Section 6, although we have cut the line about robust sensititivity as that needs more validation in more regions. Within this first effort, however, the quantitative comparisons of AIRS-OE and AIRS-V6 against radiosondes indicate that a qualitative phrase like "compares well" is reasonable. We thought we'd end the paper where the initial motivation for this effort began – a better horizontal resolution for temperature and water vapor, which when combined with cloud data, we expect to be useful in many kinds of targeted studies.

Additionally, we found a programming error that led to an undercount of successful AIRS-OE retrievals matched to sondes. This programming error has been corrected.

We thank the reviewer for perceptive comments and good questions.

---

## Author Comment (AC2) · 30 Nov 2017

In response to some concerns of the first reviewer, this revision of our paper has a significant change to the retrieval algorithm in that we removed the initial, "tentative" retrieval of cloud optical depth ($\tau_{cld}$) prior to the simultaneous retrieval of $\tau_{cld}$, $T_{cldtop}$, $r_{eff}$, $T_{sfc}$, $T_{atm}$, $H_2O$, $O_3$ and $CO_2$. Changes to results were comparatively minor, but we thought significant enough to modify our algorithm. Conclusions are for the most part unchanged. Figures have been updated, as well as quantitative comparisons to radiosondes, AIRS-V6 and CloudSat/CALIPSO. Please see our response to the first reviewer for more details.

*Overall assessment. This manuscript tackles one of the most fundamental issues in the field of hyperspectral IR remote sensing, that is the problem of clouds in an inversion scheme. The text is very well written and provides a way forward to the advancement of the inversion problem. All assumptions in the treatment of the cloud a priori are well explained. This lays out the foundation for a constructive conversation on the future directions that IR retrieval developers may want to consider when it comes to address the presence of clouds in an inversion methodology. I suggest this manuscript to be published, pending minor corrections as outlined below.*

*Minor comments: 1. The authors should add that, besides not needing absorption and scattering in the forward model, another advantage of cloud clearing is that it also does not require a formal cloud geophysical a priori and its error co-variance; it is computationally fast; it enables full column retrievals. A limitation, besides assuming constant water vapor in the cluster of FOVs, comes from also assuming uniform surface properties over the cluster of FOVs, which can be challenging over coastal regions.*

We agree that there are significant differences in the benefits/drawbacks of single-footprint vs. cloud-cleared spectra, which for brevity we did not go into. Instead, we discuss what we thought the most important difference, namely the horizontal resolution, which shows up most significantly in the water vapor and relative humidity.

*2. Could the authors explain in few sentences how their work differs from the cited existing methods employed for direct use of cloudy infrared spectra in atmospheric retrievals (Liu, 2009, Blackwell 2005 and Kulawik, 2006a).*

We hesitate to do this as differences between AIRS-OE and the Liu, Blackwell and Kulawik methods would be fairly lengthy even if described cogently, and our paper is already quite long. We thought readers would be better served by going to those papers directly, and more important (and perhaps more interesting from the reader's perspective) to contrast AIRS-OE with the AIRS operational retrieval, AIRS-V6.

*3. Can the authors say anything about using a monthly averaged surface emissivity, is this a robust approximation for a given day of the month? Also, have they evaluated the inter-annual variability in the data set to derive a rough assessment of how good of an approximation is to apply the 2003 climatology to the year before? Have they*

*attempted a comparison with the existing surface emissivity regression solution of AIRS v6?*

In Sec. 3.2.3, we note that the emissivity is not retrieved in this version of AIRS-OE, but we noted an emissivity retrieval may happen in future versions. We have added a note about this in comparing AIRS-OE to AIRS-V6 in the introduction. In Sec. 3.7.2, we also note that using a fixed emissivity is an error source that we have not yet quantified. The use of a fixed emissivity can be an admitted deficiency of the algorithm in its current state. At this stage of AIRS-OE development, it may be premature to make a robust evaluation of the emissivity as its calculation may change in future versions.

*If not, can they say few words about their motivation to replace the AIRS v6 method? Also, for completeness, they should include this as part of the list of differences with respect to AIRS v6 on page 2 -3*

This effort was a proof-of-concept not meant to "replace" AIRS-V6, but rather to provide a means where studies, in particular those of water vapor, can be done on the native resolution of the AIRS infrared observations.  In the introductory comparison with AIRS-V6, we've noted that AIRS-OE does not retrieve emissivity, although that might change.

*4. Do the MODIS cloud parameters come with a formally computed error covariance? Otherwise, what is the source for this?*

The MODIS Level 2 files used have quality flags, but do not have an error covariance provided; we do not use the MODIS data directly in the retrieval, but rather a weighted average of them. We have used ad hoc error covariances in the retrieval, as described in Table 1.

*5. Was there a specific geophysical regime where the use of the 9.6 micron band was problematic, for example the tropical region, or desert areas? The broader scientific community would benefit from a more detailed explanation of what the sentence "We therefore retrieve O3 only as an "interferent" gas within the 14 µm CO2 region, and avoid the 9.6 µm band" means. Did the authors only use channels in the CO2 band for the retrieval of ozone? If not, can they explain more?*

In Sec. 3.5, we have expanded our explanation of why we didn't use the 9.6 micron band, note that we are not currently retrieving ozone as a "primary" product, and caution against using our O3 retrievals for scientific investigation. In the conclusions section, we briefly discuss extended the retrieval to retrieve ozone using the 9.6 micron band.

*6. Section 3.6 Is the approximation of leaving all other variables fixed taken into account in the retrieval equation, for example, similarly to what is done in AIRS v6? Also, in the list of differences from the AIRS v6 approach, the authors stated that this is a simultaneous retrieval approach. Can they more fairly say that this is a two-step sequential approach?*

We note in Sec. 3.7.2 the error from parameters not retrieved, and how future versions will incorporate an estimate from these. As noted earlier, and for reasons we have written in the response to the first reviewer, we have revised our procedures to go directly into the simultaneous retrieval.

*7. Also about section 3.6, if in within the page limits, can the authors say few words on the convergence criteria, besides mentioning the reference to Bowmann et al, 2006? Are those the same as outlined later in the text? If yes, for clarity, the authors should refer the reader to that section.*

We've put some numbers on convergence and exit criteria in the solver, but in the interests of brevity, we refer the reader to the Bowman and Moré papers.

*8. What enters in the measurement noise covariance term, only instrument noise? Can the authors specify this?*

Currently only the diagonal of the instrument noise is in the measurement error covariance term. We discuss expanding this term in future version to include correlated noise and random error from the forward model.

*9. As an alternative method to the median of the medians, the authors should try the methodology described in Nalli et al., "Validation of satellite sounder environmental data records: Application to the Cross-track Infrared Microwave Sounder Suite", JGR, 2013, where each layer statistic is weighted by the mean water vapor quantity in the layer. This method was also used in Tobin et al., 2005, JGR and seems to be a well established methodology in the satellite retrieval field.*

At this stage of AIRS-OE development (and validation), the difficulty was not so much from statistics skewing for $H_2O$ in dry regions, as discussed by Nalli et al. in their Sec 3.1.2. Given that our retrievals were quite different than those on cloud-cleared radiances, the problem was how to combine retrieval statistics when there are widely differing results because of cloud cover, and still make a fair a comparison to AIRS-V6. An important consideration was how to avoid skewing the statistics so that (near-)clear scenes are would not be overrepresented. The "median of the medians" approach was developed to succintly indicate how well the AIRS-OE retrieval was doing without getting too granular, and see how it compares to the current AIRS-V6 retrieval. As the retrieval matures, and more data are analyzed, we'll reconsider our validation using something more akin to Nalli et al.

*10. Summary and discussion section. How are the authors planning to extend this retrieval method to multi-layer cloud retrievals. Does the MODIS retrieval output contain multi-layer clouds that can be used as a priori?*

The MODIS data can tell us if there are clouds at different cloud-top temperatures in the scene, but their vertical extents, particularly at night, are non-trivial issues we would need to work on. We briefly mention using weather prediction for this in the conclusion, as did DeSouza-Machado et al. (2017).

*11. Could the authors also consider comparing their cloud retrieval products to the AIRS v6 cloud retrieved products?*

We have looked at comparisons between the Kahn et al. (2014) optical estimation retrieval of cirrus (Kahn2014), which uses the AIRS-V6 cloud-top temperature as a priori, against the AIRS-OE retrieval. (Note that the Kahn2014 retrieval does not include liquid water clouds.) We include a comparison figure below. We did not include it for brevity's sake as we felt that validation of the temperature and water vapor (and relative humidity) was more needed, as their increased horizontal resolution was what was new about this work (at least compared to AIRS-V6). We also feel that a detailed cloud comparison needs to go well beyond a simple comparison such as we show below, thus we did not include in the manuscript. We did include the comparison with CloudSat/CALIPSO as an initial check against a completely different instrument suite, however.

Comparisons for Granule 44, Sept 6, 2002 are shown below, colored by the AIRS-OE averaging kernels (AKs) for each parameter. Results for $\tau_{eff}$ are well correlated for AKs close to unity, but the AIRS-OE results are biased low (~25%) compared to Kahn2014. Cloud-top temperature shows mostly good agreement for higher AKs. There is more disagreement for effective radius, $r_{eff}$, but we note that the areas where they differ greatly (Kahn2014 $r_{eff}$ > ~60 microns in the lower right), almost all the matchups have Kahn2014 quality flags of 2 ("Do Not Use"). The $r_{eff}$ retrieval is sensitive to the "tilt" of the spectrum, so some of the $r_{eff}$ difference may be related to the spectral range of channels used; the upper range of the Kahn2014 channels is ~1200 cm$^{-1}$, while AIRS-OE goes to ~1600 cm$^{-1}$. This will require more investigation, and we will continue to compare results as the AIRS-OE retrieval evolves, and we process more data.

[Figure]

*Technical comments: 1. The broader science community might not be familiar with what "L1b" data are. The author should either keep using the "Level 1" definition for radiance data or explain what the "b" stands for.*

We've added some language clarifying this in the first paragraph of the introduction.

Additionally, we found a programming error that led to an undercount of successful AIRS-OE retrievals matched to sondes (Figures 14 and 15 in the revised draft). This error has been corrected.

We thank the reviewer for perceptive comments and good questions.